# On the Roles of LLMs in Planning: Embedding LLMs into Planning Graphs

## Abstract

Plan synthesis aims to generate a course of actions or policies to transit given initial states to goal states, provided domain models that could be designed by experts or learnt from training data or interactions with the world. Intrigued by the claims of emergent planning capabilities in large language models (LLMs), works have been proposed to investigate the planning effectiveness of LLMs, without considering any utilization of off-the-shelf planning techniques in LLMs. In this paper, we aim to further study the insight of the planning capability of LLMs by investigating the roles of LLMs in off-the-shelf planning frameworks. To do this, we investigate the effectiveness of embedding LLMs into one of the well-known planning frameworks, graph-based planning, proposing a novel LLMs-based planning framework with LLMs embedded in two levels of planning graphs, i.e., mutual constraints generation level and constraints solving level. We empirically exhibit the effectiveness of our proposed framework in various planning domains.

## 1 Introduction

Plan synthesis aims to generate a course of actions or policies to transit given initial states to goal states, provided domain models that could be designed by experts or learnt from training data [1] or interactions with the world [10, 9]. It is a time- and space-consuming open issue in the planning community [6]. Intrigued by the claims of emergent planning capabilities in large language models (LLMs), works have been proposed to investigate the planning effectiveness of LLMs, without considering any utilization of off-the-shelf planning techniques in LLMs [15]. As demonstrated by [15], even in a seemingly simple common-sense domain like Blocksworld that humans usually find easy to solve, LLMs are evaluated to be quite ineffective in planning autonomously.

An interesting result shown by [15] is when taking the solution generated by LLMs, which is incorrect, as a seed plan to be repaired by an off-the-shelf planner, e.g., LPG [4], a significant improvement in search steps can be attained over the result when an empty plan provided as a seed plan for the planner. This indicates that LLMs can indeed provide some helpful information (e.g., in some sense of heuristics) for planning, even though they cannot solve planning problems solely. Inspired by the result of loosely using plans generated by LLMs as seed plans, we are curious if it is possible to "dig" more helpful information from LLMs to assist planning deeply, e.g., by inserting LLMs into planning frameworks. By doing this, we aim to answer the question: **what roles can be played exactly by LLMs in planning?** Indeed, there have been attempts to explore off-the-shelf planning techniques to help LLMs solving planning problems [12]. Similar to [15], they only view planners as black-boxes without deepening the integration of LLMs in planning frameworks.

To do this, we investigate the effectiveness of embedding LLMs into one of the well-known planning frameworks, graph-based planning [2]. We propose a novel LLMs-based planning framework with LLMs embedded in two phases of the planning framework (namely `LLMs4Plan`). The first phase is

to propose promising actions in "action-levels" of the planning graph using LLMs. The second phase
is to propose non-mutual action sets using LLMs when backtracking the planning graph. Note that
the two phases correspond to two critical steps that influence the efficiency and effectiveness in graph
planning.

*For example, as shown in Figure 1(a), there could be a large number of actions in "action-level 1" when expanding "state-label 0" with Graphplan [2]. We aim to exploit LLMs to help select a small subset of promising actions, e.g., $\{a_1, a_2, a_3\}$ are selected in Figure 1(a). In Figure 1(b), when backtracking from "state-level K" that includes goals, there could be a large number of candidate sets of actions to be explored (e.g., "action set 1", "action set 2", "action set 3") — actions in each candidate set are not mutually exclusive with each other (two actions are mutually **exclusive** if they are not allowed to be executed at the same time, e.g., actions "pick up object A" and "put down object A" are mutually exclusive). It is particularly time-consuming to search all of the valid candidate sets based on* mutual constraints *in each action-level. We expect LLMs are capable of selecting a small number of candidate sets to be backtracked, e.g., only "action set 1" is selected to be backtracked by LLMs as shown in Figure 1(b).*

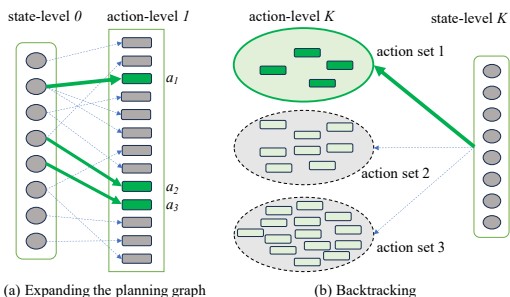

(a) Expanding the planning graph    (b) Backtracking

Figure 1: Two critical steps in graph planning

Specifically, in `LLMs4Plan`, for each action-level $i$, we first automatically generate prompts based on *propositions* in state-level $i - 1$, *goals* and *domain models*, and feed the prompts to LLMs to select actions for generating action-level $i$. After that, when backtracking from the last state-level $K$ in the expanded planning graph, which includes the goal, we automatically generate prompts based on propositions in state-level $K$, propositions in the initial state (i.e., state-level 0), mutual constraints in action-level $K$, and *domain models*, and feed the prompts to LLMs to select action sets for backtracking. We embed the above two components into one of the well-known off-the-shelf graph planners, Graphplan [2]. We study the effectiveness of different cases of adding or removing the above one or two components in Graphplan to see the significance of roles LLMs play in the graph planning framework.

Through this study, we provide new clues for how to *deeply* embed LLMs into off-the-shelf planning frameworks, i.e., first identifying critical steps (generally time-consuming ones) in specific planning frameworks, and then designing proper prompt generation to be embedded into the frameworks. We verify that soly relying on LLMs to do planning is far from a good option, while leveraging LLMs to help deal with some critical steps in the graph planning framework is possible.

## 2   Problem Formulation

In this work we consider classical planning problems specified in the form of STRIPS [3]. Similar ideas can be extended into more expressive planning language such as PDDL [5]. Let $\mathcal{L}$ be a set of atoms, each of which is composed of a predicate with zero or more parameters (e.g., *clean(room)* is an atom indicating *room* is *clean*). A STRIPS domain is composed of a set of action models $\mathcal{A}$, each of which is a quadruple $\langle a, \text{PRE}(a), \text{ADD}(a), \text{DEL}(a)\rangle$, where $a$ is an action name with zero or more parameters, $\text{PRE}(a) \subseteq \mathcal{L}$ is a precondition list indicating the conditions under which $a$ can be applied, $\text{ADD}(a) \subseteq \mathcal{L}$ is an adding list and $\text{DEL}(a) \subseteq \mathcal{L}$ is a deleting list indicating the effects of $a$. Let $\mathcal{R}$ be a set of propositions, which are instances of atoms in $\mathcal{L}$. We define a planning problem as $\mathcal{P} = \langle \mathcal{R}, s_0, g, \mathcal{A} \rangle$, where $s_0 \subseteq \mathcal{R}$ is an initial state and $g \subseteq \mathcal{R}$ is a goal. A solution $\pi$ to the planning problem is a sequence of actions that transit initial state $s_0$ to goal $g$. An intuitive example of our planning problem is as shown below.

*Suppose we would like to clean a bedroom using a vacuum which is placed in a tool room. We can formulate the problem $\mathcal{P} = \langle \mathcal{R}, s_0, g, \mathcal{A} \rangle$ in the form of STRIPS (note that we assume there is no parameter for each predicate and action for simplicity since there is only one tool, one bedroom and one toolroom). The set of propositions $\mathcal{R}$ is represented by $\mathcal{R} = \{dirty(), toolroom(), clean(), bedroom()\}$. Initial state $s_0$ is represented by $s_0$ = {dirty(), toolroom()}, which indicates the "bedroom" is*

*dirty, and the tool "vacuum" is in the tool room (i.e., "toolroom"). The goal $g$ is represented by*
*$g = \{clean(), toolroom()\}$, which indicates the "bedroom" is clean, and the tool "vacuum" is back*
*to the tool room. The set of action models $\mathcal{A}$ is represented as follows:*

| Action | Preconditions | Effects |
|---|---|---|
| $vacuum()$ | $dirty(), bedroom()$ | $clean(), \neg dirty()$ |
| $move2tr()$ | $bedroom()$ | $toolroom(), \neg bedroom()$ |
| $move2br()$ | $toolroom()$ | $bedroom(), \neg toolroom()$ |

*Action $vacuum()$ aims to vacuuming "bedroom", the preconditions of which are "bedroom" is*
*dirty and the vacuum-cleaner is in "bedroom". The effects of $vacuum()$ are adding $clean()$ to*
*the state where $vacuum()$ is executed, indicating "bedroom" is clean, and deleting $dirty()$ (i.e.,*
*$\neg dirty()$) from the state, indicating "bedroom" is not dirty anymore. Action $move2tr()$ aims*
*to move the vacuum-cleaner to "toolroom", the precondition of which is $bedroom()$ indicating*
*the vacuum-cleaner is in "bedroom". The effects are adding $toolroom()$ indicating the vacuum-*
*cleaner is in "toolroom", and deleting $bedroom()$, indicating the vacuum-cleaner is not in "bed-*
*room". Similarly, action $move2br$ aims to move the vacuum-cleaner to "bedroom", the precondi-*
*tion of which is $toolroom()$, indicating the vacuum-cleaner is in "toolroom". The effects are the*
*vacuum-cleaner is adding $bedroom()$, indicating the vacuum-cleaner is in "bedroom", deleting*
*$toolroom()$, indicating the vacuum-cleaner is not in "toolroom". A solution $\pi$ to the problem $\mathcal{P}$ is*
*$move2br(), vacuum(), move2tr()$.*

## 3   Our `LLMs4Plan` approach

An overview of our `LLMs4Plan` approach is shown in Algorithm **??**. In Step 3, the pruning possibility
$\kappa_i$ is decreased as the exponent $i$ increasing. In Step 5, we expand planning graph $PG^r$ with one
more level using LLMs to prune actions based on pruning possibility $\kappa_i$ and planning problem
$\mathcal{P}$. In Steps 7, if goal $g$ is not included by the last state-level in $PG^r$, i.e., $Satisfied(g, PG^r)$ is
false, we continue to Step 4. In Step 8, we build a set of mutual constraints $\mathcal{C}$ based on $PG^r$, i.e.,
$buildConstraints(PG^r)$. In Step 9, we sort sets of actions based on constraints $\mathcal{C}$ using LLMs,
i.e., $sortActionsLLMs(PG^r, \mathcal{C}$. In Step 10, we search solution $\pi$ based on the sorted action sets $\mathbb{A}$
using depth-first search. In the following subsections, we will address our `LLMs4Plan` in detail.

---

**Algorithm 1** An overview of our `LLMs4Plan`

**Input:** Planning problem $\mathcal{P}$, pruning possibility $\kappa_0$
**Output:** Solution $\pi$

1:  $PG^r = \emptyset$
2:  **for** $i = 1$ to $N$ **do**
3:    $\kappa_i = (\kappa_0)^i$, $k = 1$
4:    **while** $k < K$ **do**
5:      $PG^r \leftarrow expandGraphLLMs(PG^r, \mathcal{P}, \kappa_i)$
6:      $k = k + 1$
7:      if $Satisfied(g, PG^r) = false$, then **continue**
8:      $\mathcal{C} = buildConstraints(PG^r)$
9:      $\mathbb{A} = sortActionsLLMs(PG^r, \mathcal{C})$
10:     $\pi = depthFirstSearch(\mathbb{A}, PG^r)$
11:     if $\pi \neq$ Failure, then **return** $\pi$
12:   **end while**
13: **end for**
14: **return** Failure

---

### 3.1   Building Planning Graphs with LLMs

A planning graph $PG^r$ is the search space for a relaxed version of the planning problem, an
intuitive framework of which is shown in Figure 5. It alternates layers of ground literals and
actions. "Square" nodes at action-level $i + 1$ indicate actions that might be possible to be exe-

121 cuted in state $s_i$. Maintenance actions indicate dump operators that keep literals unchanged be-
122 tween state-levels $i$ and $i + 1$. "Black circle" nodes at state-level $i$ indicate literals that might
123 possibly be true at time $i$. Edges between state-level $i$ and action-level $i$ indicate literals in
124 state-level $i$ are preconditions of actions in action-level $i$, while edges between action-level $i$ and
125 state-level $i + 1$ indicate literals in state-level $i + 1$ are adding or deleting effects of actions in
126 action-level $i$. The nodes in the first state-level indicate literals that are true in initial state $s_0$.

127

128 The procedure of building the planning graph
129 with LLMs (i.e., *buildGraphLLMs*) based on the
130 given planning problem $\mathcal{P} = \langle \mathcal{R}, s_0, g, \mathcal{A} \rangle$ is as
131 follows:

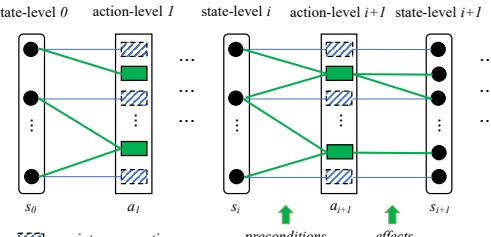

132     1. All propositions in $s_0$ and negation of
133        propositions in $\mathcal{R} - s_0$ are added into
134        state-level 0.

135     2. All actions in $\mathcal{A}$, whose preconditions
136        are satisfied in state-level 0 and **se-**
137        **lected by LLMs**, are added into action-
138        level 1; a maintenance action corre-
139        sponding to each proposition in state-
140        level 0 is added into action-level 1.

Figure 2: The framework of a planning graph

141     3. The propositions added or deleted by
142        actions in action-level 1 are added into state-level 1; and all propositions in state-level 0 are
143        added into state-level 1 as well (i.e., which is done by the maintenance action).

144     4. We repeat steps 1-3 by increasing state-
145        level 0 to 1 (or $i$ to $i+1$) until all propo-
146        sitions in goal $g$ are included by state-
147        level $k$.

148 In Step 5, we use LLMs to help select actions to build the planning
149 graph. Note that in classical graph-based planning [2], all of the ac-
150 tions whose preconditions are satisfied will be added into the action-level.
151 We design the prompt to consult LLMs as shown in Figure 3,
152 where "$\langle$domain$\rangle$", "$\langle$initial state$\rangle$", "$\langle$goal$\rangle$", "$\langle$proposition set$\rangle$",
153 and "$\langle$candidate actions$\rangle$" are action models $\mathcal{A}$, initial state $s_0$, goal
154 $g$, the set of propositions $\mathcal{R}$ and all of the candidate actions whose
155 preconditions are satisfied in $s_0$. The text in BLUE is the prompt
156 used to guide LLMs to select actions. "$\langle$example of output format$\rangle$"
157 is used to guide LLMs to output actions in the desired format, e.g.,
158 "move '?from': 'rooma', '?to': 'roomb'".

```
<domain>
<initial state>
<goal>
<proposition set>
<candidate actions>
Analyze each predicate in the state one by one
to list a smallest subset in the following format
from above candidate actions list that have the
potential to achieve the goal state.
<example of output format>
```

Figure 3: The prompt for prun-
ing actions

159 ## 3.2 Building Mutual Constraints

160 Due to the satisfaction of action models being relaxed, actions and/or
161 states in action-levels or state-labels may be inconsistent, i.e., there
162 may be some actions mutually exclusive in action-levels, or some literals mutually exclusive in
163 state-levels. As shown in Figure 4, there are three types of mutual exclusion constraints among
164 actions. Specifically, two actions at the same action-level are mutex, if they satisfy the following
165 conditions:

166     • An effect of one negates an effect of the other, which is called *inconsistent effects*.

167     • One deletes a precondition of the other, which is called *interference*.

168     • They have mutually exclusive preconditions, which is called *Competing needs*.

169 Otherwise they do not interfere with each other, i.e., both may appear in a solution plan. Two literals
170 at the same state-level are mutex if one is the negation of the other, or all ways of achieving them are
171 pairwise mutex, namely *inconsistent support*.

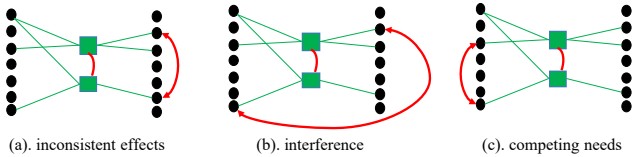

(a). inconsistent effects      (b). interference      (c). competing needs

Figure 4: Mutual exclusion of actions

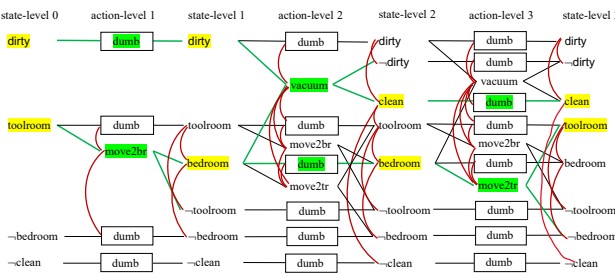

Figure 5: An example of planning graph and mutual constraints indicated in RED arcs

An example planning graph corresponding to Example 1 is as shown in Figure 5. Action *vacuum* is mutually exclusive with action *dumb* for *toolroom* at *action-level 2* since *vacuum*'s precondition *bedroom* is mutually exclusive with *toolroom* at *state-level 1*.

### 3.3   Sort Action Sets with LLMs and Search Solutions

After we build a set of constraints in Step 8 of Algorithm 1, we use off-the-shelf procedure presented in [2] to compute candidate action sets such that there are no conflicts (i.e., satisfying the constraints $\mathcal{C}$) among actions in each action set. After that, in Step 9, we consult LLMs to sort the action sets by designing the prompts as shown in Figure 6, which is similar to the prompt shown in Figure 3 except the command in BLUE. After we get the sorted action sets $\mathbb{A}$, in Step 10, we conduct the dept-first search procedure as done in [2] by giving the priority of action sets based on the sorted action sets in $\mathbb{A}$.

```
<domain>
<initial state>
<goal>
<proposition set>
<candidate actions>
Reorder the above candidate actions and
output them in the following format. The
actions that can directly reach the goal state
are ranked higher.
<example of output format>
```

Figure 6: The prompt for sorting action sets

## 4   Experiment

### 4.1   Experimental Setup

In the experiment, we evaluate `LLMs4Plan` in ten planning domains with different scenarios, including gripper, miconic, logistics, movie, blocks, satellite, zenotravel, driverlog, woodworking and openstacks. Ten problems are randomly selected for each domain. The specific scenarios and sizes are described in Appendix A.1.

To demonstrate the effectiveness of our `LLMs4Plan` approach, we designed five sets of comparison experiments. The methods were implemented using Python. We compared our `LLMs4Plan` approach with four other methods, which are listed below:

- **GP**: It is the graph-based planning algorithm mentioned above. We implement the traditional graph planning algorithm as the most important baseline for comparison. We directly provide *domain.pddl* and *problem.pddl* to the planner for solving.

- **GPT-3.5**: We simply construct and splice the contents of *domain.pddl* and *problem.pddl* directly. We then add the necessary command prompts to form a complete prompt into GPT3.5 and command the model to solve the problem directly.

- **GPT-4**: The process is the same as GPT-3.5.

- `LLMs4Plan`-**GPT3.5**: When we are expanding the hierarchy or backtracking, we are faced with a candidate action selection with LLMs. We guide the LLMs to select a minimal subset of actions from them and utilize this subset of actions for the next algorithmic operations, where LLMs are specifically GPT-3.5.
- `LLMs4Plan`-**GPT4**: Replaced the LLM model with GPT4, otherwise same as `LLMs4Plan`-GPT3.5.

## 4.2 Experimental Metrics

In the experimental framework described, we employed three distinct metrics to assess the efficacy of various methodologies: the problem-solving success rate, the cumulative count of expansion actions and the node count for backtracking in Depth-First Search (DFS).

**Problem-solving success rate**. The solvability of a problem is a crucial metric in assessing planning problems. All approaches are required to generate a sequence of actions that is sufficient to solve the problem, and only if the problem can be transferred from the initial state to the goal state through this sequence of actions can the corresponding problem be considered to be successfully solved. Furthermore, we have established an upper bound on the depth of the problem-solving process. The optimal length of the action sequence for the test problem is known to us. Should the solution obtained surpass this predetermined depth, it signifies the inability of the method to successfully ascertain the optimal action path for this particular problem. Setting an upper bound on the depth of the problem-solving process serves the purpose of not only requiring the planner to solve problems but also demanding that it does so more efficiently. This ensures that the output action sequences are more concise and accurate, minimizing the occurrence of redundant actions.

**Total number of expansion actions**. In the GP algorithm, the expansion of actions at each layer is a fundamental process, and the number of these expansions serves as a vital metric. Under the premise of preserving effective actions, fewer expansions result in a reduced count of mutually exclusive action pairs and subsequently fewer branches in the deep search phase of backtracking, thereby enhancing efficiency. Consequently, we compute the average total number of action expansions per layer across all problems, applying different methods within various domains, as a significant metric for comparison.

**Number of nodes for backtracking DFS**. This metric serves as the cornerstone for validating our optimization efforts, as the DFS during backtracking accounts for the majority of the computational load in the GP algorithm, overshadowing the forward expansion phase. Particularly when dealing with increasing expansion depths, the exponentially growing number of DFS poses the most significant challenge for GP algorithms in tackling large-scale problems or complex solution sequences. We primarily utilize this metric to ascertain which method truly enhances the efficiency of the planning process.

Regarding the number of nodes for backtracking DFS, our analysis was confined to data from the GP and `LLMs4Plan`-GPT4 methods, primarily for two reasons. Firstly, the metric is relevant only in scenarios where the problem is successfully solved; failed solutions do not yield countable data. Consequently, we excluded `LLMs4Plan`-GPT3.5 from our statistical analysis due to its comparatively lower success rate. Secondly, these metrics are inherently calculable within the GP framework alone. Hence, directly solving problems using GPT-3.5 and GPT-4 precludes the possibility of gathering this data, as these methods operate outside the GP framework.

## 4.3 Experimental Results

We present the success rates in Table 1, depict the pruning effects of action expansion for LLM on GP in Figure 7, and showcase experimental results in Table 2 comparing our approach to traditional GP algorithms in terms of the number of nodes for backtracking DFS metrics, along with relevant ablation studies.

**Ablation Experiment**: We conducted four ablation experiments to ascertain the effectiveness of forward pruning and backward sorting, detailed as follows:

1. `LLMs4Plan`: This method involves both forward pruning and backward sorting.
2. `LLMs4Plan`-**unsorted**: Here, we implement pruning without sorting.

Table 1: Success rate results. In the table, each row corresponds to a distinct domain, while each column represents a separate approach or method. The values presented within the table indicate the success rate for each combination of domain and approach, with these rates quantified on a scale ranging from 0 to 1.

| | GPT-3.5 | GPT-4 | GP | LLMs4Plan-GPT3.5 | LLMs4Plan-GPT4 |
|---|---|---|---|---|---|
| **gripper** | 0.00 | 0.60 | 0.70 | 0.00 | **1.00** |
| **miconic** | 0.10 | 0.50 | 0.60 | 0.10 | **1.00** |
| **logistics** | 0.20 | 0.60 | 0.60 | 0.20 | **1.00** |
| **movie** | **1.00** | **1.00** | **1.00** | **1.00** | **1.00** |
| **blocks** | 0.10 | 0.70 | 0.60 | 0.30 | **1.00** |
| **satellite** | 0.00 | 0.50 | 0.90 | 0.10 | **1.00** |
| **zenotravel** | 0.20 | 0.60 | 0.90 | 0.20 | **1.00** |
| **driverlog** | 0.00 | 0.10 | 0.90 | 0.20 | **1.00** |
| **woodworking** | 0.90 | 0.90 | 0.70 | **1.00** | **1.00** |
| **openstacks** | 0.10 | 0.20 | **1.00** | 0.20 | **1.00** |

3. LLMs4Plan-**unpruned**: In this approach, sorting is used, but not pruning.

4. **GP**: This method involves neither pruning nor sorting.

Table 2: In the table, each row corresponds to a distinct domain, while each column represents a group of ablation experiments. The values presented within the table indicate the number of nodes for backtracking DFS. Both pruning and sorting effectively enhance search efficiency, leading to a substantial reduction in the number of nodes required for searching. Generally, pruning tends to be slightly more effective than sorting.

| | LLMs4Plan | LLMs4Plan-**unsorted** | LLMs4Plan-**unpruned** | GP |
|---|---|---|---|---|
| **gripper** | **6839** | 11294 | 4850376 | 8486698 |
| **miconic** | **11891** | 47863 | 445145 | 2018484 |
| **logistics** | **59** | 85 | 1226 | 1261250 |
| **movie** | **975163** | 1211830 | **975163** | 10869160 |
| **blocks** | **4572** | 7272 | 83129 | 1205223 |
| **satellite** | **46619** | 94811 | 67167049 | 88779785 |
| **zenotravel** | **1548** | 12166 | 839527 | 2259283 |
| **driverlog** | **574** | 1916 | 58311 | 1579486 |
| **woodworking** | **49** | 3924 | 48553 | 114502 |
| **openstacks** | **107** | 409 | 13577 | 24267 |

## 4.4 Experimental Analysis

**Analysis of the success rate of planning**: From Table 1, several conclusions can be drawn. GPT3.5 exhibits competence primarily in resolving simple problems with short action sequence lengths, such as in the **movie** domain, while its success rates are notably low in other domains. Consequently, it struggles to enhance the capabilities of GP algorithms. Conversely, GPT4 demonstrates substantial improvements in abilities compared to GPT3.5, particularly in reasoning skills and decision-making involving long action sequences. With the enhanced reasoning and commonsense capabilities of GPT4, GP shows an enhanced success rate in certain domains. We observe instances of failure in GP, attributed to the inclusion of partially corrupt data during testing. Specifically, we introduce a proportion of corrupted data by randomly removing action preconditions and effects propositions from domain files. These instances have varying impacts across different domains, particularly affecting traditional GP algorithms reliant on the completeness of domain files. We detail the influence of random removal on success rates in Table 3. For each layer of GP expansion, the provision of action preconditions and effects propositions is essential. However, in the case of LLM-augmented GP algorithms, LLMs4Plan is capable of making rational action decisions even in the presence of missing action propositions, thereby aiding in the completion of current planning tasks. Consequently, our algorithm exhibits greater robustness in terms of success rates compared to traditional GP approaches. The specific settings for the robustness experiments are detailed more extensively in section A.2 of the appendix.

Table 3: This table illustrates experiments on the robustness of missing action predicates. In the table, each row corresponds to a distinct domain. Each column in the table represents the proportion of predicates we removed. A higher proportion indicates a greater amount of missing information, posing increased difficulty for the planner to solve the problem. The values in the table represent the success rates of GP in solving the problems. In the majority of domains, as the proportion of deleted predicates increases, the success rate of GP planning decreases. Overall, this indicates that GP exhibits poor robustness to missing action predicates.

|  | **10%** | **20%** | **30%** | **40%** | **50%** |
|---|---|---|---|---|---|
| **gripper** | 0.40 | 0.87 | 0.73 | 0.80 | 0.67 |
| **miconic** | 0.60 | 0.60 | 0.60 | 0.80 | 0.40 |
| **logistics** | 0.20 | 0.80 | 0.80 | 0.67 | 0.60 |
| **movie** | 1.00 | 1.00 | 1.00 | 1.00 | 0.60 |
| **blocks** | 1.00 | 0.26 | 0.07 | 0.27 | 0.47 |
| **satellite** | 0.60 | 0.73 | 1.00 | 0.86 | 0.87 |
| **zenotravel** | 0.93 | 0.93 | 1.00 | 0.93 | 0.80 |
| **driverlog** | 0.80 | 1.00 | 1.00 | 1.00 | 1.00 |
| **woodworking** | 1.00 | 0.73 | 0.40 | 0.27 | 0.40 |
| **openstacks** | 1.00 | 1.00 | 1.00 | 1.00 | 1.00 |

Upon examining the generated action sequences, we observed that although GPT4 achieves a certain level of success in solving problems, the action sequences it produces tend to be longer compared to those generated by GP alone. By integrating GPT4 with graph planning, `LLMs4Plan` can effectively generate more optimal action sequences.

**Analysis of search efficiency**: Besides planning success rates, our method significantly improves search efficiency compared to GP algorithms. This enhancement is evident from Table 2, where, among problems with successful planning outputs, we drastically reduce the cost of search nodes, achieving an exponential level of optimization. So, **how does the LLM-augmented GP method enhance search efficiency?**

Through in-depth analysis of experimental cases, we identify two main aspects of optimization:

1. During forward expansion, LLM efficiently and effectively prunes the expansion actions, leading to varying degrees of stable reduction in the total number of expanded actions and mutually exclusive actions. Consequently, the computational load of forward expansion decreases correspondingly.

2. During the depth-first search backtracking process, LLM prioritizes searching closer to the set of planning solutions, accelerating the attainment of planning solutions and saving time by avoiding ineffective searches.

We provide an example from the 'logistics' domain to illustrate our analysis in Figure 7, where we compare the number of expansion actions before and after LLM pruning. The application of LLM for pruning demonstrates significant efficacy across all layers. In addition, we have provided further experimental results and analysis on the total number of mutually exclusive actions and expanded actions in section A.3 of the supplementary materials.

For pruning, the greatest risk is removing necessary actions, rendering the problem unsolvable. In our experiments, we observed instances where LLM prunes crucial actions in certain layers, resulting in the inability to obtain effective solutions. However, we introduced pruning probabilities to ensure algorithm completeness. Experimental results demonstrate that although the process of correcting LLM's erroneous pruning behavior through pruning probabilities may introduce additional expansion and search steps, the cumulative cost of these search steps remains significantly lower than the cost of solely using GP algorithms to solve problems. The results presented in Table 2 compare the outcomes of our method `LLMs4Plan`, which ensures completeness, with those of GP algorithms.

**Analysis of ablation experiments**: Table 2 reveals that both pruning and sorting contribute to enhanced search efficiency, with their combination amplifying this effect. Notably, pruning appears slightly more effective than sorting. This is likely because LLM, while pruning, also organizes the remaining actions logically. In contrast, sorting may lead to minor errors due to the multitude of actions and lengthy text. In this regard, we require further optimization of natural language processing

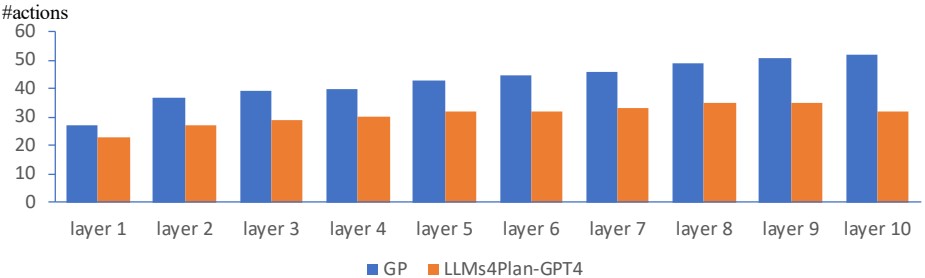

#actions

Figure 7: An example of action pruning. Both the GP and `LLMs4Plan`-GPT4 methods expanded through 10 layers. The horizontal axis represents the layer number, with lower numbers indicating proximity to the initial state and higher numbers nearing the goal state. The vertical axis shows the count of expanded actions, including both domain-specific actions and numerous empty actions, characteristic of the graph planning algorithm. This implies a high pruning ratio for genuinely effective actions. LLMs prune almost every layer of expansion actions and the data in the table also contains many empty actions.

techniques tailored for handling extremely long texts to enhance our framework's capability in solving more complex problems. The experiments also indicate that LLM tends not to prioritize empty actions in graph planning, favoring their later arrangement. This aligns with our analysis suggesting that prioritizing non-empty actions is more productive, as a layer without any action is essentially redundant.

**Analysis of the advantages and disadvantages of `LLMs4Plan`**: Upon analyzing examples where solutions failed, we observed that GPT4 is more prone to pruning errors at deeper expansion levels. This results in the discarding of effective actions, thereby unnecessarily increasing the expansion layers and hindering problem resolution. We attribute this to two primary factors. Firstly, as the expansion level deepens, both the predicate set and the candidate action set expand, leading to increasingly lengthy input prompts. This prolonged text can cause GPT-4 to gradually lose track of previous information, resulting in decision-making errors. Secondly, the nature of the predicate set in graph planning diverges from traditional planning's current state representation. This discrepancy impairs LLM's ability to accurately analyze the predicate set, leading to the erroneous elimination of effective actions. LLM lacks capacity to analyze complex predicate set combinations.

Our analysis of additional failure examples indicates that graph planning excels in efficiently handling numerous non-mutually exclusive actions in parallel, due to its ability to group these actions within the same layer. However, its limitation becomes apparent in scenarios requiring the execution of highly complex and extremely long action sequences. If a problem's optimal solution sequences are lengthy, the planning graph must be expanded considerably deeper. Despite effective pruning, this does not resolve the issue of exponential complexity growth in backtracking DFS caused by increased depth.

## 5   Conclusion and Future Work

Our comparative experiments in multiple domains demonstrated the efficacy of our `LLMs4Plan` in significantly enhancing the problem-solving capabilities of graph planning algorithms. Notably, `LLMs4Plan` boosts not just the success rate of problem resolution but also markedly enhances search efficiency and substantially reduces computational complexity. The runtime of `LLMs4Plan` is currently hindered by multiple LLMs calls. While our method requires multiple LLMs calls, it provides substantially improved results. There are also various ways to enhance runtime performance like using smaller LLMs like Llama [14] or distilling LLMs' knowledge into a smaller model [13, 7, 11]. Those are interesting avenues for future research. Instead of leveraging LLMs to assist planning, it would also be possible to study acquring action models [16] and more planning frameworks [8] with the help of LLMs.

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

# A   Appendix / supplemental material

## A.1   Description of Testing Domains

In the experiment, we evaluate `LLMs4Plan` in ten planning domains with different scenarios, including gripper, miconic, logistics, movie, blocks, satellite, zenotravel, driverlog, woodworking and openstacks, which are:

- **Gripper**: Tasks utilize an existing robot arm to move objects between rooms, with between 6 and 20 objects in the scene.
- **Miconic**: Tasks uses elevators to serve guests between floors and help them reach the floor they want to go to, with between 6 and 50 objects in the scene.
- **Logistics**: One of the classic logistics problems. Transportation of items between different locations in different cities using trucks and airplanes, with between 10 and 30 objects in the scenario.
- **Movie**: Simulate some simple behaviors while watching a movie with between 45 and 155 objects in the scene.
- **Blocks**: As one of the most classic planning problems, it involves manipulating various blocks on a table using a robotic arm to achieve specific goals. The number of blocks in the scenario ranges between 20 and 40.
- **Satellite**: Satellites equipped with various instruments perform different tasks in space. The total number of entities, including instruments and satellites, ranges from 20 to 40.
- **Zenotravel**: This field involves the problem of passengers traveling between different cities by airplane, including the management of aircraft fuel. The total number of entities in the scenario ranges from 20 to 30.
- **Driverlog**: Different truck drivers need to coordinate the transportation of goods between platforms using trucks. The total number of entities in the scenario ranges from 20 to 30.
- **Woodworking**: A carpenter needs to operate various machines, wooden boards, and components in the workshop to complete processing tasks. The total number of entities in the scenario ranges from 30 to 40.
- **Openstacks**: In an assembly line, the corresponding number of products are produced based on order tasks. The total number of entities in the scenario ranges from 10 to 30.

## A.2   Robustness Experimental Setup

In the robustness experiments in Table 3, we randomly delete a certain proportion (e.g., 10%, 20%, 30%, 40%, 50%) of action preconditions and effects propositions (referred to collectively as conditions) from the domain files and assess whether the GP algorithm can produce correct solutions. We conduct five sets of experiments for each proportion in every domain, with each experiment tested three times to obtain the average success rate. In addition to the previously defined criteria, failures also include cases where the GP algorithm may become stuck in a loop or program deadlock due to missing predicates. Therefore, we set an extended time threshold. If the solving process exceeds this threshold, we consider it as a failed solution. Timeliness in planning is crucial; if a problem remains unsolved for several times the normal solving duration, the planner becomes impractical.

## A.3   Additional Experiments

In this section, our primary focus lies in elucidating the fundamental reasons behind the efficiency of our algorithm compared to the GP algorithm. In this section, our primary objective is to delve into

the fundamental reasons underlying the efficiency of our algorithm in contrast to the GP algorithm. To commence, we will elucidate two evaluation metrics: the total number of expansion actions and the total number of mutually exclusive actions.

**Total number of expansion actions**. In the GP algorithm, the expansion of actions at each layer is a fundamental process, and the number of these expansions serves as a vital metric. Under the premise of preserving effective actions, fewer expansions result in a reduced count of mutually exclusive action pairs and subsequently fewer branches in the deep search phase of backtracking, thereby enhancing efficiency. Consequently, we compute the average total number of action expansions per layer across all problems, applying different methods within various domains, as a significant metric for comparison.

**Total number of mutually exclusive actions**. Mutually exclusive actions are generated from the set of candidate expansion actions based on many different mutually exclusive conditions. The critical aspect of this metric lies in the fact that a lower total number of mutually exclusive actions translates into significant computational time savings. This efficiency is evident both during the generation of the mutually exclusive action set in the forward expansion phase and in the filtering of candidate actions based on this set during the backtracking process.

We conducted statistical analysis on relevant metrics for several domains, and the results are presented in Table 4. The data indicates that both the total number of expansion actions and mutually exclusive actions experience a consistent decline with LLM integration. This suggests a reduction in the computational effort required for forward expansion. Overall, these data corroborate the conclusion drawn from our experimental analysis regarding the efficiency of `LLMs4Plan`.

Table 4: LLM pruning results. In the table, each column signifies a distinct domain. It displays the statistical outcomes of the two methods across two indicators. Smaller numbers in the table denote lower computational resource usage and greater effectiveness.

| Domain | Expansion Actions | | Mutex Actions | |
|---|---|---|---|---|
| | GP | LLMs4Plan | GP | LLMs4Plan |
| **gripper** | 157 | **128** | 1656 | **1179** |
| **miconic** | 181 | **176** | 792 | **668** |
| **logistics** | 294 | **167** | 1726 | **161** |
| **movie** | 401 | **308** | 7 | 7 |

**A.4 Experimental Case Presentation**

In this section, we will present a specific case to illustrate how we integrate LLM into the GP algorithm. This particular case is drawn from a problem in logistics domain. In our demonstration, we will present three sets of actions at each layer involved in addressing this problem. The first set of actions pertains to graph planning expansion and backtracking, denoted as the GP-ACTION-SET. The second set of actions corresponds to the expansion process of the Large Language Model (LLM), designated as the LLM-EP-ACTION-SET. The third set of actions relates to the backtracking process of the LLM, termed as the LLM-BP-ACTION-SET.

The domain of logistics is described in PDDL language as follows:

```
(define (domain logistics-strips)
(:requirements :strips)
(:predicates (OBJ ?obj) (TRUCK ?truck) (LOCATION ?loc) (AIRPLANE ?airplane)
(CITY ?city) (AIRPORT ?airport) (at ?obj ?loc) (in ?obj1 ?obj2) (in-city ?obj ?city))

(:action LOAD-TRUCK
:parameters (?obj ?truck ?loc)
:precondition (and (OBJ ?obj) (TRUCK ?truck) (LOCATION ?loc) (at ?truck ?loc) (at ?obj
?loc))
:effect (and (not (at ?obj ?loc)) (in ?obj ?truck)))

(:action LOAD-AIRPLANE
:parameters (?obj ?airplane ?loc)
:precondition (and (OBJ ?obj) (AIRPLANE ?airplane) (LOCATION ?loc) (at ?obj ?loc) (at
?airplane ?loc))
:effect (and (not (at ?obj ?loc)) (in ?obj ?airplane)))

(:action UNLOAD-TRUCK
:parameters (?obj ?truck ?loc)
:precondition (and (OBJ ?obj) (TRUCK ?truck) (LOCATION ?loc) (at ?truck ?loc) (in ?obj
?truck))
:effect (and (not (in ?obj ?truck)) (at ?obj ?loc)))

(:action UNLOAD-AIRPLANE
:parameters (?obj ?airplane ?loc)
:precondition (and (OBJ ?obj) (AIRPLANE ?airplane) (LOCATION ?loc) (in ?obj
?airplane) (at ?airplane ?loc))
:effect (and (not (in ?obj ?airplane)) (at ?obj ?loc)))

(:action DRIVE-TRUCK
:parameters (?truck ?loc-from ?loc-to ?city)
:precondition (and (TRUCK ?truck) (LOCATION ?loc-from) (LOCATION ?loc-to) (CITY
?city) (at ?truck ?loc-from) (in-city ?loc-from ?city) (in-city ?loc-to ?city))
:effect (and (not (at ?truck ?loc-from)) (at ?truck ?loc-to)))

(:action FLY-AIRPLANE
:parameters (?airplane ?loc-from ?loc-to)
:precondition (and (AIRPLANE ?airplane) (AIRPORT ?loc-from) (AIRPORT ?loc-to) (at
?airplane ?loc-from))
:effect (and (not (at ?airplane ?loc-from)) (at ?airplane ?loc-to))))
```

The problem of logistics is described in PDDL language as follows:

```
(define (problem logistics-02)
(:domain logistics-strips)
(:objects
a0
c0 c1
t0 t1
l00 l01 l10 l11
p0 p1
)
(:init
(AIRPLANE a0) (CITY c0) (CITY c1) (TRUCK t0) (TRUCK t1) (LOCATION l00) (in-city
l00 c0) (LOCATION l01) (in-city l01 c0) (LOCATION l10) (in-city l10 c1) (LOCATION
l11) (in-city l11 c1) (AIRPORT l00) (AIRPORT l10) (at a0 l00) (OBJ p0) (OBJ p1) (at t0
l00) (at t1 l10) (at p0 l01) )
(:goal
(and (at p0 l11) ) )
)
```

468

We will demonstrate the sets of the aforementioned three types of actions layer by layer, and annotate the final output of the planning solution with bold fonts. The first layer represents the initial expansion action layer, and so on up to the final goal layer, totaling 10 layers.From this specific case, we can observe the following points:

1. Although LLM undergoes pruning, the correct planning solution always exists within the LLM-EP-ACTION-SET, with the removed actions being redundant.

2. The size of the LLM-EP-ACTION-SET is always smaller than that of the GP-ACTION-SET.

3. In the LLM-BP-ACTION-SET, LLM consistently positions the correct planning solution actions towards the front among most other actions. Although not always the first, they are generally placed near the beginning.

479      1. Layer 1
480          GP-ACTION-SET

481

```
DRIVE-TRUCK {'?truck': 't1', '?loc-from': 'l10', '?loc-to': 'l11', '?city': 'c1'}
{('at', 't1', 'l11')}
NoOp {} {('LOCATION', 'l10')}
NoOp {} {('in-city', 'l01', 'c0')}
NoOp {} {('CITY', 'c1')}
NoOp {} {('OBJ', 'p1')}
NoOp {} {('AIRPLANE', 'a0')}
NoOp {} {('LOCATION', 'l11')}
NoOp {} {('at', 't1', 'l10')}
NoOp {} {('in-city', 'l00', 'c0')}
DRIVE-TRUCK {'?truck': 't1', '?loc-from': 'l10', '?loc-to': 'l10', '?city': 'c1'}
{('at', 't1', 'l10')}
DRIVE-TRUCK {'?truck': 't0', '?loc-from': 'l00', '?loc-to': 'l00', '?city': 'c0'}
{('at', 't0', 'l00')}
FLY-AIRPLANE {'?airplane': 'a0', '?loc-from': 'l00', '?loc-to': 'l00'} {('at',
'a0', 'l00')}
NoOp {} {('LOCATION', 'l01')}
NoOp {} {('in-city', 'l11', 'c1')}
NoOp {} {('AIRPORT', 'l10')}
DRIVE-TRUCK {'?truck': 't0', '?loc-from': 'l00', '?loc-to': 'l01', '?city':
'c0'} {('at', 't0', 'l01')}
NoOp {} {('CITY', 'c0')}
FLY-AIRPLANE {'?airplane': 'a0', '?loc-from': 'l00', '?loc-to': 'l10'} {('at',
'a0', 'l10')}
NoOp {} {('at', 'a0', 'l00')}
NoOp {} {('LOCATION', 'l00')}
NoOp {} {('TRUCK', 't1')}
NoOp {} {('OBJ', 'p0')}
NoOp {} {('TRUCK', 't0')}
NoOp {} {('AIRPORT', 'l00')}
NoOp {} {('at', 't0', 'l00')}
NoOp {} {('in-city', 'l10', 'c1')}
NoOp {} {('at', 'p0', 'l01')}
```

482          LLM-EP-ACTION-SET

```
DRIVE-TRUCK {'?truck': 't1', '?loc-from': 'l10', '?loc-to': 'l11', '?city': 'c1'}
{('at', 't1', 'l11')}
NoOp {} {('LOCATION', 'l10')}
NoOp {} {('in-city', 'l01', 'c0')}
NoOp {} {('CITY', 'c1')}
NoOp {} {('OBJ', 'p1')}
NoOp {} {('AIRPLANE', 'a0')}
NoOp {} {('LOCATION', 'l11')}
NoOp {} {('at', 't1', 'l10')}
NoOp {} {('in-city', 'l00', 'c0')}
NoOp {} {('LOCATION', 'l01')}
NoOp {} {('in-city', 'l11', 'c1')}
NoOp {} {('AIRPORT', 'l10')}
DRIVE-TRUCK {'?truck': 't0', '?loc-from': 'l00', '?loc-to': 'l01', '?city':
'c0'} {('at', 't0', 'l01')}
NoOp {} {('CITY', 'c0')}
NoOp {} {('at', 'a0', 'l00')}
NoOp {} {('LOCATION', 'l00')}
NoOp {} {('TRUCK', 't1')}
NoOp {} {('OBJ', 'p0')}
NoOp {} {('TRUCK', 't0')}
NoOp {} {('AIRPORT', 'l00')}
NoOp {} {('at', 't0', 'l00')}
NoOp {} {('in-city', 'l10', 'c1')}
NoOp {} {('at', 'p0', 'l01')}
```

484    LLM-BP-ACTION-SET

```
DRIVE-TRUCK {'?truck': 't1', '?loc-from': 'l10', '?loc-to': 'l11', '?city': 'c1'}
{('at', 't1', 'l11')}
DRIVE-TRUCK {'?truck': 't0', '?loc-from': 'l00', '?loc-to': 'l01', '?city':
'c0'} {('at', 't0', 'l01')}
NoOp {} {('LOCATION', 'l10')}
NoOp {} {('in-city', 'l01', 'c0')}
NoOp {} {('CITY', 'c1')}
NoOp {} {('OBJ', 'p1')}
NoOp {} {('AIRPLANE', 'a0')}
NoOp {} {('LOCATION', 'l11')}
NoOp {} {('at', 't1', 'l10')}
NoOp {} {('in-city', 'l00', 'c0')}
NoOp {} {('LOCATION', 'l01')}
NoOp {} {('in-city', 'l11', 'c1')}
NoOp {} {('AIRPORT', 'l10')}
NoOp {} {('CITY', 'c0')}
NoOp {} {('at', 'a0', 'l00')}
NoOp {} {('LOCATION', 'l00')}
NoOp {} {('TRUCK', 't1')}
NoOp {} {('OBJ', 'p0')}
NoOp {} {('TRUCK', 't0')}
NoOp {} {('AIRPORT', 'l00')}
NoOp {} {('at', 't0', 'l00')}
NoOp {} {('in-city', 'l10', 'c1')}
NoOp {} {('at', 'p0', 'l01')}
```

486    2. Layer 2
487       GP-ACTION-SET

```
NoOp {} {('LOCATION', 'l10')}
NoOp {} {('at', 'p0', 'l01')}
NoOp {} {('TRUCK', 't0')}
FLY-AIRPLANE {'?airplane': 'a0', '?loc-from': 'l00', '?loc-to': 'l10'} {('at',
'a0', 'l10')}
NoOp {} {('LOCATION', 'l11')}
NoOp {} {('TRUCK', 't1')}
NoOp {} {('LOCATION', 'l01')}
NoOp {} {('OBJ', 'p0')}
FLY-AIRPLANE {'?airplane': 'a0', '?loc-from': 'l00', '?loc-to': 'l00'} {('at',
'a0', 'l00')}
DRIVE-TRUCK {'?truck': 't0', '?loc-from': 'l00', '?loc-to': 'l00', '?city': 'c0'}
{('at', 't0', 'l00')}
NoOp {} {('at', 't0', 'l00')}
DRIVE-TRUCK {'?truck': 't1', '?loc-from': 'l10', '?loc-to': 'l11', '?city': 'c1'}
{('at', 't1', 'l11')}
NoOp {} {('AIRPORT', 'l10')}
DRIVE-TRUCK {'?truck': 't1', '?loc-from': 'l11', '?loc-to': 'l10', '?city': 'c1'}
{('at', 't1', 'l10')}
DRIVE-TRUCK {'?truck': 't1', '?loc-from': 'l10', '?loc-to': 'l10', '?city': 'c1'}
{('at', 't1', 'l10')}
NoOp {} {('in-city', 'l11', 'c1')}
NoOp {} {('in-city', 'l00', 'c0')}
NoOp {} {('LOCATION', 'l00')}
DRIVE-TRUCK {'?truck': 't0', '?loc-from': 'l01', '?loc-to': 'l01', '?city': 'c0'}
{('at', 't0', 'l01')}
NoOp {} {('at', 'a0', 'l00')}
NoOp {} {('CITY', 'c0')}
NoOp {} {('AIRPLANE', 'a0')}
NoOp {} {('in-city', 'l10', 'c1')}
LOAD-TRUCK {'?obj': 'p0', '?truck': 't0', '?loc': 'l01'} {('in', 'p0', 't0')}
NoOp {} {('CITY', 'c1')}
DRIVE-TRUCK {'?truck': 't0', '?loc-from': 'l00', '?loc-to': 'l01', '?city': 'c0'}
{('at', 't0', 'l01')}
NoOp {} {('OBJ', 'p1')}
NoOp {} {('AIRPORT', 'l00')}
NoOp {} {('at', 't0', 'l01')}
NoOp {} {('at', 't1', 'l11')}
NoOp {} {('in-city', 'l01', 'c0')}
NoOp {} {('at', 't1', 'l10')}
DRIVE-TRUCK {'?truck': 't0', '?loc-from': 'l01', '?loc-to': 'l00', '?city': 'c0'}
{('at', 't0', 'l00')}
DRIVE-TRUCK {'?truck': 't1', '?loc-from': 'l11', '?loc-to': 'l11', '?city': 'c1'}
{('at', 't1', 'l11')}
```

LLM-EP-ACTION-SET

```
NoOp {} {('LOCATION', 'l10')}
NoOp {} {('at', 'p0', 'l01')}
NoOp {} {('TRUCK', 't0')}
FLY-AIRPLANE {'?airplane': 'a0', '?loc-from': 'l00', '?loc-to': 'l10'} {('at',
'a0', 'l10')}
NoOp {} {('LOCATION', 'l11')}
NoOp {} {('TRUCK', 't1')}
NoOp {} {('LOCATION', 'l01')}
NoOp {} {('OBJ', 'p0')}
NoOp {} {('at', 't0', 'l00')}
DRIVE-TRUCK {'?truck': 't1', '?loc-from': 'l10', '?loc-to': 'l11', '?city': 'c1'}
{('at', 't1', 'l11')}
NoOp {} {('AIRPORT', 'l10')}
NoOp {} {('in-city', 'l11', 'c1')}
NoOp {} {('in-city', 'l00', 'c0')}
NoOp {} {('LOCATION', 'l00')}
NoOp {} {('at', 'a0', 'l00')}
NoOp {} {('CITY', 'c0')}
NoOp {} {('AIRPLANE', 'a0')}
NoOp {} {('in-city', 'l10', 'c1')}
LOAD-TRUCK {'?obj': 'p0', '?truck': 't0', '?loc': 'l01'} {('in', 'p0', 't0')}
NoOp {} {('CITY', 'c1')}
NoOp {} {('OBJ', 'p1')}
NoOp {} {('AIRPORT', 'l00')}
NoOp {} {('at', 't0', 'l01')}
NoOp {} {('at', 't1', 'l11')}
NoOp {} {('in-city', 'l01', 'c0')}
NoOp {} {('at', 't1', 'l10')}
DRIVE-TRUCK {'?truck': 't0', '?loc-from': 'l01', '?loc-to': 'l00', '?city': 'c0'}
{('at', 't0', 'l00')}
```

491        LLM-BP-ACTION-SET

```
LOAD-TRUCK {'?obj': 'p0', '?truck': 't0', '?loc': 'l01'} {('in', 'p0', 't0')}
DRIVE-TRUCK {'?truck': 't1', '?loc-from': 'l10', '?loc-to': 'l11', '?city': 'c1'}
{('at', 't1', 'l11')}
FLY-AIRPLANE {'?airplane': 'a0', '?loc-from': 'l00', '?loc-to': 'l10'} {('at',
'a0', 'l10')}
DRIVE-TRUCK {'?truck': 't0', '?loc-from': 'l01', '?loc-to': 'l00', '?city': 'c0'}
{('at', 't0', 'l00')}
NoOp {} {('LOCATION', 'l10')}
NoOp {} {('at', 'p0', 'l01')}
NoOp {} {('TRUCK', 't0')}
NoOp {} {('LOCATION', 'l11')}
NoOp {} {('TRUCK', 't1')}
NoOp {} {('LOCATION', 'l01')}
NoOp {} {('OBJ', 'p0')}
NoOp {} {('at', 't0', 'l00')}
NoOp {} {('AIRPORT', 'l10')}
NoOp {} {('in-city', 'l11', 'c1')}
NoOp {} {('in-city', 'l00', 'c0')}
NoOp {} {('LOCATION', 'l00')}
NoOp {} {('at', 'a0', 'l00')}
NoOp {} {('CITY', 'c0')}
NoOp {} {('AIRPLANE', 'a0')}
NoOp {} {('in-city', 'l10', 'c1')}
NoOp {} {('CITY', 'c1')}
NoOp {} {('OBJ', 'p1')}
NoOp {} {('AIRPORT', 'l00')}
NoOp {} {('at', 't0', 'l01')}
NoOp {} {('at', 't1', 'l11')}
NoOp {} {('in-city', 'l01', 'c0')}
NoOp {} {('at', 't1', 'l10')}
```

3. Layer 3
    GP-ACTION-SET

NoOp {} {('AIRPORT', 'l10')}
NoOp {} {('in', 'p0', 't0')}
NoOp {} {('at', 'a0', 'l10')}
NoOp {} {('in-city', 'l11', 'c1')}
UNLOAD-TRUCK {'?obj': 'p0', '?truck': 't0', '?loc': 'l01'} {('at', 'p0', 'l01')}
NoOp {} {('OBJ', 'p1')}
NoOp {} {('OBJ', 'p0')}
DRIVE-TRUCK {'?truck': 't1', '?loc-from': 'l11', '?loc-to': 'l10', '?city': 'c1'} {('at', 't1', 'l10')}
DRIVE-TRUCK {'?truck': 't1', '?loc-from': 'l10', '?loc-to': 'l11', '?city': 'c1'} {('at', 't1', 'l11')}
NoOp {} {('at', 't0', 'l00')}
NoOp {} {('LOCATION', 'l00')}
NoOp {} {('AIRPLANE', 'a0')}
FLY-AIRPLANE {'?airplane': 'a0', '?loc-from': 'l10', '?loc-to': 'l00'} {('at', 'a0', 'l00')}
NoOp {} {('at', 'p0', 'l01')}
NoOp {} {('CITY', 'c0')}
NoOp {} {('LOCATION', 'l10')}
NoOp {} {('in-city', 'l00', 'c0')}
DRIVE-TRUCK {'?truck': 't0', '?loc-from': 'l00', '?loc-to': 'l01', '?city': 'c0'} {('at', 't0', 'l01')}
DRIVE-TRUCK {'?truck': 't0', '?loc-from': 'l01', '?loc-to': 'l01', '?city': 'c0'} {('at', 't0', 'l01')}
DRIVE-TRUCK {'?truck': 't1', '?loc-from': 'l11', '?loc-to': 'l11', '?city': 'c1'} {('at', 't1', 'l11')}
DRIVE-TRUCK {'?truck': 't0', '?loc-from': 'l00', '?loc-to': 'l00', '?city': 'c0'} {('at', 't0', 'l00')}
**DRIVE-TRUCK {'?truck': 't0', '?loc-from': 'l01', '?loc-to': 'l00', '?city': 'c0'} {('at', 't0', 'l00')}**
NoOp {} {('LOCATION', 'l01')}
LOAD-TRUCK {'?obj': 'p0', '?truck': 't0', '?loc': 'l01'} {('in', 'p0', 't0')}
FLY-AIRPLANE {'?airplane': 'a0', '?loc-from': 'l00', '?loc-to': 'l10'} {('at', 'a0', 'l10')}
NoOp {} {('at', 't0', 'l01')}
NoOp {} {('at', 'a0', 'l00')}
NoOp {} {('AIRPORT', 'l00')}
NoOp {} {('in-city', 'l01', 'c0')}
NoOp {} {('CITY', 'c1')}
FLY-AIRPLANE {'?airplane': 'a0', '?loc-from': 'l00', '?loc-to': 'l00'} {('at', 'a0', 'l00')}
NoOp {} {('LOCATION', 'l11')}
NoOp {} {('at', 't1', 'l11')}
NoOp {} {('TRUCK', 't0')}
FLY-AIRPLANE {'?airplane': 'a0', '?loc-from': 'l10', '?loc-to': 'l10'} {('at', 'a0', 'l10')}
NoOp {} {('at', 't1', 'l10')}
DRIVE-TRUCK {'?truck': 't1', '?loc-from': 'l10', '?loc-to': 'l10', '?city': 'c1'} {('at', 't1', 'l10')}
NoOp {} {('TRUCK', 't1')}
NoOp {} {('in-city', 'l10', 'c1')}

LLM-EP-ACTION-SET

```
NoOp {} {('AIRPORT', 'l10')}
NoOp {} {('in', 'p0', 't0')}
NoOp {} {('at', 'a0', 'l10')}
NoOp {} {('in-city', 'l11', 'c1')}
NoOp {} {('OBJ', 'p1')}
NoOp {} {('OBJ', 'p0')}
DRIVE-TRUCK {'?truck': 't1', '?loc-from': 'l10', '?loc-to': 'l11', '?city': 'c1'}
{('at', 't1', 'l11')}
NoOp {} {('at', 't0', 'l00')}
NoOp {} {('LOCATION', 'l00')}
NoOp {} {('AIRPLANE', 'a0')}
NoOp {} {('at', 'p0', 'l01')}
NoOp {} {('CITY', 'c0')}
NoOp {} {('LOCATION', 'l10')}
NoOp {} {('in-city', 'l00', 'c0')}
DRIVE-TRUCK {'?truck': 't0', '?loc-from': 'l01', '?loc-to': 'l00', '?city':
'c0'} {('at', 't0', 'l00')}
NoOp {} {('LOCATION', 'l01')}
LOAD-TRUCK {'?obj': 'p0', '?truck': 't0', '?loc': 'l01'} {('in', 'p0', 't0')}
FLY-AIRPLANE {'?airplane': 'a0', '?loc-from': 'l00', '?loc-to': 'l10'} {('at',
'a0', 'l10')}
NoOp {} {('at', 't0', 'l01')}
NoOp {} {('at', 'a0', 'l00')}
NoOp {} {('AIRPORT', 'l00')}
NoOp {} {('in-city', 'l01', 'c0')}
NoOp {} {('CITY', 'c1')}
NoOp {} {('LOCATION', 'l11')}
NoOp {} {('at', 't1', 'l11')}
NoOp {} {('TRUCK', 't0')}
NoOp {} {('at', 't1', 'l10')}
NoOp {} {('TRUCK', 't1')}
NoOp {} {('in-city', 'l10', 'c1')}
```

LLM-BP-ACTION-SET

```
DRIVE-TRUCK {'?truck': 't1', '?loc-from': 'l10', '?loc-to': 'l11', '?city': 'c1'}
{('at', 't1', 'l11')}
DRIVE-TRUCK {'?truck': 't0', '?loc-from': 'l01', '?loc-to': 'l00', '?city':
'c0'} {('at', 't0', 'l00')}
LOAD-TRUCK {'?obj': 'p0', '?truck': 't0', '?loc': 'l01'} {('in', 'p0', 't0')}
FLY-AIRPLANE {'?airplane': 'a0', '?loc-from': 'l00', '?loc-to': 'l10'} {('at',
'a0', 'l10')}
NoOp {} {('AIRPORT', 'l10')}
NoOp {} {('in', 'p0', 't0')}
NoOp {} {('at', 'a0', 'l10')}
NoOp {} {('in-city', 'l11', 'c1')}
NoOp {} {('OBJ', 'p1')}
NoOp {} {('OBJ', 'p0')}
NoOp {} {('at', 't0', 'l00')}
NoOp {} {('LOCATION', 'l00')}
NoOp {} {('AIRPLANE', 'a0')}
NoOp {} {('at', 'p0', 'l01')}
NoOp {} {('CITY', 'c0')}
NoOp {} {('LOCATION', 'l10')}
NoOp {} {('in-city', 'l00', 'c0')}
NoOp {} {('LOCATION', 'l01')}
NoOp {} {('at', 't0', 'l01')}
NoOp {} {('at', 'a0', 'l00')}
NoOp {} {('AIRPORT', 'l00')}
NoOp {} {('in-city', 'l01', 'c0')}
NoOp {} {('CITY', 'c1')}
NoOp {} {('LOCATION', 'l11')}
NoOp {} {('at', 't1', 'l11')}
NoOp {} {('TRUCK', 't0')}
NoOp {} {('at', 't1', 'l10')}
NoOp {} {('TRUCK', 't1')}
NoOp {} {('in-city', 'l10', 'c1')}
```

499

500   4. Layer 4
501       GP-ACTION-SET

```
NoOp {} {('in-city', 'l10', 'c1')}
NoOp {} {('LOCATION', 'l10')}
NoOp {} {('in-city', 'l11', 'c1')}
FLY-AIRPLANE {'?airplane': 'a0', '?loc-from': 'l10', '?loc-to': 'l00'} {('at',
'a0', 'l00')}
NoOp {} {('TRUCK', 't1')}
NoOp {} {('LOCATION', 'l11')}
NoOp {} {('at', 't0', 'l00')}
NoOp {} {('AIRPORT', 'l00')}
DRIVE-TRUCK {'?truck': 't0', '?loc-from': 'l00', '?loc-to': 'l01', '?city': 'c0'}
{('at', 't0', 'l01')}
NoOp {} {('OBJ', 'p0')}
NoOp {} {('in-city', 'l01', 'c0')}
NoOp {} {('in-city', 'l00', 'c0')}
NoOp {} {('at', 'a0', 'l10')}
DRIVE-TRUCK {'?truck': 't1', '?loc-from': 'l10', '?loc-to': 'l11', '?city': 'c1'}
{('at', 't1', 'l11')}
NoOp {} {('CITY', 'c1')}
FLY-AIRPLANE {'?airplane': 'a0', '?loc-from': 'l10', '?loc-to': 'l10'} {('at',
'a0', 'l10')}
UNLOAD-TRUCK {'?obj': 'p0', '?truck': 't0', '?loc': 'l01'} {('at', 'p0', 'l01')}
DRIVE-TRUCK {'?truck': 't0', '?loc-from': 'l01', '?loc-to': 'l00', '?city': 'c0'}
{('at', 't0', 'l00')}
NoOp {} {('at', 't1', 'l11')}
DRIVE-TRUCK {'?truck': 't0', '?loc-from': 'l00', '?loc-to': 'l00', '?city': 'c0'}
{('at', 't0', 'l00')}
NoOp {} {('OBJ', 'p1')}
NoOp {} {('at', 't0', 'l01')}
NoOp {} {('TRUCK', 't0')}
DRIVE-TRUCK {'?truck': 't1', '?loc-from': 'l11', '?loc-to': 'l10', '?city': 'c1'}
{('at', 't1', 'l10')}
NoOp {} {('AIRPORT', 'l10')}
LOAD-TRUCK {'?obj': 'p0', '?truck': 't0', '?loc': 'l01'} {('in', 'p0', 't0')}
NoOp {} {('CITY', 'c0')}
FLY-AIRPLANE {'?airplane': 'a0', '?loc-from': 'l00', '?loc-to': 'l00'} {('at',
'a0', 'l00')}
NoOp {} {('LOCATION', 'l01')}
NoOp {} {('at', 'a0', 'l00')}
NoOp {} {('at', 'p0', 'l01')}
DRIVE-TRUCK {'?truck': 't0', '?loc-from': 'l01', '?loc-to': 'l01', '?city': 'c0'}
{('at', 't0', 'l01')}
UNLOAD-TRUCK {'?obj': 'p0', '?truck': 't0', '?loc': 'l00'} {('at', 'p0',
'l00')}
NoOp {} {('in', 'p0', 't0')}
NoOp {} {('AIRPLANE', 'a0')}
NoOp {} {('at', 't1', 'l10')}
DRIVE-TRUCK {'?truck': 't1', '?loc-from': 'l11', '?loc-to': 'l11', '?city': 'c1'}
{('at', 't1', 'l11')}
NoOp {} {('LOCATION', 'l00')}
FLY-AIRPLANE {'?airplane': 'a0', '?loc-from': 'l00', '?loc-to': 'l10'} {('at',
'a0', 'l10')}
DRIVE-TRUCK {'?truck': 't1', '?loc-from': 'l10', '?loc-to': 'l10', '?city': 'c1'}
{('at', 't1', 'l10')}
```

502

503     LLM-EP-ACTION-SET

```
NoOp {} {('in-city', 'l10', 'c1')}
NoOp {} {('LOCATION', 'l10')}
NoOp {} {('in-city', 'l11', 'c1')}
NoOp {} {('TRUCK', 't1')}
NoOp {} {('LOCATION', 'l11')}
NoOp {} {('at', 't0', 'l00')}
NoOp {} {('AIRPORT', 'l00')}
NoOp {} {('OBJ', 'p0')}
NoOp {} {('in-city', 'l01', 'c0')}
NoOp {} {('in-city', 'l00', 'c0')}
NoOp {} {('at', 'a0', 'l10')}
DRIVE-TRUCK {'?truck': 't1', '?loc-from': 'l10', '?loc-to': 'l11', '?city': 'c1'}
{('at', 't1', 'l11')}
NoOp {} {('CITY', 'c1')}
DRIVE-TRUCK {'?truck': 't0', '?loc-from': 'l01', '?loc-to': 'l00', '?city': 'c0'}
{('at', 't0', 'l00')}
NoOp {} {('at', 't1', 'l11')}
NoOp {} {('OBJ', 'p1')}
NoOp {} {('at', 't0', 'l01')}
NoOp {} {('TRUCK', 't0')}
NoOp {} {('AIRPORT', 'l10')}
LOAD-TRUCK {'?obj': 'p0', '?truck': 't0', '?loc': 'l01'} {('in', 'p0', 't0')}
NoOp {} {('CITY', 'c0')}
NoOp {} {('LOCATION', 'l01')}
NoOp {} {('at', 'a0', 'l00')}
NoOp {} {('at', 'p0', 'l01')}
UNLOAD-TRUCK {'?obj': 'p0', '?truck': 't0', '?loc': 'l00'} {('at', 'p0',
'l00')}
NoOp {} {('in', 'p0', 't0')}
NoOp {} {('AIRPLANE', 'a0')}
NoOp {} {('at', 't1', 'l10')}
NoOp {} {('LOCATION', 'l00')}
FLY-AIRPLANE {'?airplane': 'a0', '?loc-from': 'l00', '?loc-to': 'l10'} {('at',
'a0', 'l10')}
```

504

505    LLM-BP-ACTION-SET

DRIVE-TRUCK {'?truck': 't1', '?loc-from': 'l10', '?loc-to': 'l11', '?city': 'c1'} {('at', 't1', 'l11')}
LOAD-TRUCK {'?obj': 'p0', '?truck': 't0', '?loc': 'l01'} {('in', 'p0', 't0')}
DRIVE-TRUCK {'?truck': 't0', '?loc-from': 'l01', '?loc-to': 'l00', '?city': 'c0'} {('at', 't0', 'l00')}
**UNLOAD-TRUCK {'?obj': 'p0', '?truck': 't0', '?loc': 'l00'} {('at', 'p0', 'l00')}**
FLY-AIRPLANE {'?airplane': 'a0', '?loc-from': 'l00', '?loc-to': 'l10'} {('at', 'a0', 'l10')}
NoOp {} {('in-city', 'l10', 'c1')}
NoOp {} {('LOCATION', 'l10')}
NoOp {} {('in-city', 'l11', 'c1')}
NoOp {} {('TRUCK', 't1')}
NoOp {} {('LOCATION', 'l11')}
NoOp {} {('at', 't0', 'l00')}
NoOp {} {('AIRPORT', 'l00')}
NoOp {} {('OBJ', 'p0')}
NoOp {} {('in-city', 'l01', 'c0')}
NoOp {} {('in-city', 'l00', 'c0')}
NoOp {} {('at', 'a0', 'l10')}
NoOp {} {('CITY', 'c1')}
NoOp {} {('at', 't1', 'l11')}
NoOp {} {('OBJ', 'p1')}
NoOp {} {('at', 't0', 'l01')}
NoOp {} {('TRUCK', 't0')}
NoOp {} {('AIRPORT', 'l10')}
NoOp {} {('CITY', 'c0')}
NoOp {} {('LOCATION', 'l01')}
NoOp {} {('at', 'a0', 'l00')}
NoOp {} {('at', 'p0', 'l01')}
NoOp {} {('in', 'p0', 't0')}
NoOp {} {('AIRPLANE', 'a0')}
NoOp {} {('at', 't1', 'l10')}
NoOp {} {('LOCATION', 'l00')}

5. Layer 5
    GP-ACTION-SET

UNLOAD-TRUCK {'?obj': 'p0', '?truck': 't0', '?loc': 'l01'} {('at', 'p0', 'l01')}
NoOp {} {('AIRPORT', 'l00')}
NoOp {} {('LOCATION', 'l10')}
LOAD-TRUCK {'?obj': 'p0', '?truck': 't0', '?loc': 'l01'} {('in', 'p0', 't0')}
NoOp {} {('in', 'p0', 't0')}
NoOp {} {('in-city', 'l00', 'c0')}
NoOp {} {('CITY', 'c0')}
NoOp {} {('at', 't1', 'l11')}
NoOp {} {('LOCATION', 'l00')}
NoOp {} {('at', 'a0', 'l10')}
NoOp {} {('AIRPORT', 'l10')}
NoOp {} {('at', 'p0', 'l00')}
DRIVE-TRUCK {'?truck': 't1', '?loc-from': 'l11', '?loc-to': 'l11', '?city': 'c1'}
{('at', 't1', 'l11')}
NoOp {} {('TRUCK', 't1')}
NoOp {} {('at', 't1', 'l10')}
DRIVE-TRUCK {'?truck': 't1', '?loc-from': 'l10', '?loc-to': 'l11', '?city': 'c1'}
{('at', 't1', 'l11')}
NoOp {} {('in-city', 'l11', 'c1')}
FLY-AIRPLANE {'?airplane': 'a0', '?loc-from': 'l10', '?loc-to': 'l10'} {('at',
'a0', 'l10')}
NoOp {} {('OBJ', 'p0')}
FLY-AIRPLANE {'?airplane': 'a0', '?loc-from': 'l00', '?loc-to': 'l10'} {('at',
'a0', 'l10')}
UNLOAD-TRUCK {'?obj': 'p0', '?truck': 't0', '?loc': 'l00'} {('at', 'p0', 'l00')}
DRIVE-TRUCK {'?truck': 't0', '?loc-from': 'l01', '?loc-to': 'l00', '?city': 'c0'}
{('at', 't0', 'l00')}
LOAD-TRUCK {'?obj': 'p0', '?truck': 't0', '?loc': 'l00'} {('in', 'p0', 't0')}
DRIVE-TRUCK {'?truck': 't1', '?loc-from': 'l10', '?loc-to': 'l10', '?city': 'c1'}
{('at', 't1', 'l10')}
DRIVE-TRUCK {'?truck': 't0', '?loc-from': 'l00', '?loc-to': 'l00', '?city': 'c0'}
{('at', 't0', 'l00')}
NoOp {} {('at', 'a0', 'l00')}
NoOp {} {('at', 'p0', 'l01')}
NoOp {} {('at', 't0', 'l00')}
NoOp {} {('at', 't0', 'l01')}
NoOp {} {('LOCATION', 'l11')}
FLY-AIRPLANE {'?airplane': 'a0', '?loc-from': 'l00', '?loc-to': 'l00'} {('at',
'a0', 'l00')}
DRIVE-TRUCK {'?truck': 't0', '?loc-from': 'l01', '?loc-to': 'l01', '?city': 'c0'}
{('at', 't0', 'l01')}
NoOp {} {('in-city', 'l01', 'c0')}
NoOp {} {('OBJ', 'p1')}
FLY-AIRPLANE {'?airplane': 'a0', '?loc-from': 'l10', '?loc-to': 'l00'} {('at',
'a0', 'l00')}
NoOp {} {('in-city', 'l10', 'c1')}
NoOp {} {('AIRPLANE', 'a0')}
**LOAD-AIRPLANE {'?obj': 'p0', '?airplane': 'a0', '?loc': 'l00'} {('in', 'p0',
'a0')}**
DRIVE-TRUCK {'?truck': 't0', '?loc-from': 'l00', '?loc-to': 'l01', '?city': 'c0'}
{('at', 't0', 'l01')}
NoOp {} {('LOCATION', 'l01')}
DRIVE-TRUCK {'?truck': 't1', '?loc-from': 'l11', '?loc-to': 'l10', '?city': 'c1'}
{('at', 't1', 'l10')}
NoOp {} {('TRUCK', 't0')}
NoOp {} {('CITY', 'c1')}

510    LLM-EP-ACTION-SET

```
NoOp {} {('AIRPORT', 'l00')}
NoOp {} {('LOCATION', 'l10')}
LOAD-TRUCK {'?obj': 'p0', '?truck': 't0', '?loc': 'l01'} {('in', 'p0', 't0')}
NoOp {} {('in', 'p0', 't0')}
NoOp {} {('in-city', 'l00', 'c0')}
NoOp {} {('CITY', 'c0')}
NoOp {} {('at', 't1', 'l11')}
NoOp {} {('LOCATION', 'l00')}
NoOp {} {('at', 'a0', 'l10')}
NoOp {} {('AIRPORT', 'l10')}
NoOp {} {('at', 'p0', 'l00')}
NoOp {} {('TRUCK', 't1')}
NoOp {} {('at', 't1', 'l10')}
DRIVE-TRUCK {'?truck': 't1', '?loc-from': 'l10', '?loc-to': 'l11', '?city': 'c1'}
{('at', 't1', 'l11')}
NoOp {} {('in-city', 'l11', 'c1')}
NoOp {} {('OBJ', 'p0')}
FLY-AIRPLANE {'?airplane': 'a0', '?loc-from': 'l00', '?loc-to': 'l10'} {('at',
'a0', 'l10')}
UNLOAD-TRUCK {'?obj': 'p0', '?truck': 't0', '?loc': 'l00'} {('at', 'p0', 'l00')}
DRIVE-TRUCK {'?truck': 't0', '?loc-from': 'l01', '?loc-to': 'l00', '?city': 'c0'}
{('at', 't0', 'l00')}
NoOp {} {('at', 'a0', 'l00')}
NoOp {} {('at', 'p0', 'l01')}
NoOp {} {('at', 't0', 'l00')}
NoOp {} {('at', 't0', 'l01')}
NoOp {} {('LOCATION', 'l11')}
NoOp {} {('in-city', 'l01', 'c0')}
NoOp {} {('OBJ', 'p1')}
NoOp {} {('in-city', 'l10', 'c1')}
NoOp {} {('AIRPLANE', 'a0')}
LOAD-AIRPLANE {'?obj': 'p0', '?airplane': 'a0', '?loc': 'l00'} {('in', 'p0',
'a0')}
NoOp {} {('LOCATION', 'l01')}
NoOp {} {('TRUCK', 't0')}
NoOp {} {('CITY', 'c1')}
```

511

512     LLM-BP-ACTION-SET

DRIVE-TRUCK {'?truck': 't1', '?loc-from': 'l10', '?loc-to': 'l11', '?city': 'c1'} {('at', 't1', 'l11')}
DRIVE-TRUCK {'?truck': 't0', '?loc-from': 'l01', '?loc-to': 'l00', '?city': 'c0'} {('at', 't0', 'l00')}
UNLOAD-TRUCK {'?obj': 'p0', '?truck': 't0', '?loc': 'l00'} {('at', 'p0', 'l00')}
LOAD-TRUCK {'?obj': 'p0', '?truck': 't0', '?loc': 'l01'} {('in', 'p0', 't0')}
FLY-AIRPLANE {'?airplane': 'a0', '?loc-from': 'l00', '?loc-to': 'l10'} {('at', 'a0', 'l10')}
**LOAD-AIRPLANE {'?obj': 'p0', '?airplane': 'a0', '?loc': 'l00'} {('in', 'p0', 'a0')}**
NoOp {} {('AIRPORT', 'l00')}
NoOp {} {('LOCATION', 'l10')}
NoOp {} {('in', 'p0', 't0')}
NoOp {} {('in-city', 'l00', 'c0')}
NoOp {} {('CITY', 'c0')}
NoOp {} {('at', 't1', 'l11')}
NoOp {} {('LOCATION', 'l00')}
NoOp {} {('at', 'a0', 'l10')}
NoOp {} {('AIRPORT', 'l10')}
NoOp {} {('at', 'p0', 'l00')}
NoOp {} {('TRUCK', 't1')}
NoOp {} {('at', 't1', 'l10')}
NoOp {} {('in-city', 'l11', 'c1')}
NoOp {} {('OBJ', 'p0')}
NoOp {} {('at', 'a0', 'l00')}
NoOp {} {('at', 'p0', 'l01')}
NoOp {} {('at', 't0', 'l00')}
NoOp {} {('at', 't0', 'l01')}
NoOp {} {('LOCATION', 'l11')}
NoOp {} {('in-city', 'l01', 'c0')}
NoOp {} {('OBJ', 'p1')}
NoOp {} {('in-city', 'l10', 'c1')}
NoOp {} {('AIRPLANE', 'a0')}
NoOp {} {('LOCATION', 'l01')}
NoOp {} {('TRUCK', 't0')}
NoOp {} {('CITY', 'c1')}

6.  Layer 6

    GP-ACTION-SET

FLY-AIRPLANE {'?airplane': 'a0', '?loc-from': 'l00', '?loc-to': 'l00'} {('at', 'a0', 'l00')}
LOAD-AIRPLANE {'?obj': 'p0', '?airplane': 'a0', '?loc': 'l00'} {('in', 'p0', 'a0')}
NoOp {} {('LOCATION', 'l01')}
LOAD-TRUCK {'?obj': 'p0', '?truck': 't0', '?loc': 'l01'} {('in', 'p0', 't0')}
DRIVE-TRUCK {'?truck': 't0', '?loc-from': 'l00', '?loc-to': 'l01', '?city': 'c0'} {('at', 't0', 'l01')}
DRIVE-TRUCK {'?truck': 't0', '?loc-from': 'l01', '?loc-to': 'l00', '?city': 'c0'} {('at', 't0', 'l00')}
NoOp {} {('AIRPLANE', 'a0')}
NoOp {} {('at', 't1', 'l10')}
NoOp {} {('in-city', 'l10', 'c1')}
DRIVE-TRUCK {'?truck': 't1', '?loc-from': 'l11', '?loc-to': 'l11', '?city': 'c1'} {('at', 't1', 'l11')}
NoOp {} {('at', 't1', 'l11')}
NoOp {} {('at', 'p0', 'l01')}
NoOp {} {('in-city', 'l00', 'c0')}
NoOp {} {('at', 't0', 'l00')}
NoOp {} {('LOCATION', 'l00')}
NoOp {} {('AIRPORT', 'l10')}
NoOp {} {('at', 'p0', 'l00')}
FLY-AIRPLANE {'?airplane': 'a0', '?loc-from': 'l10', '?loc-to': 'l00'} {('at', 'a0', 'l00')}
UNLOAD-AIRPLANE {'?obj': 'p0', '?airplane': 'a0', '?loc': 'l00'} {('at', 'p0', 'l00')}
NoOp {} {('AIRPORT', 'l00')}
DRIVE-TRUCK {'?truck': 't0', '?loc-from': 'l01', '?loc-to': 'l01', '?city': 'c0'} {('at', 't0', 'l01')}
NoOp {} {('in', 'p0', 'a0')}
NoOp {} {('TRUCK', 't0')}
NoOp {} {('in', 'p0', 't0')}
NoOp {} {('at', 'a0', 'l00')}
NoOp {} {('in-city', 'l11', 'c1')}
NoOp {} {('at', 'a0', 'l10')}
NoOp {} {('LOCATION', 'l10')}
NoOp {} {('OBJ', 'p1')}
UNLOAD-TRUCK {'?obj': 'p0', '?truck': 't0', '?loc': 'l01'} {('at', 'p0', 'l01')}
NoOp {} {('in-city', 'l01', 'c0')}
LOAD-TRUCK {'?obj': 'p0', '?truck': 't0', '?loc': 'l00'} {('in', 'p0', 't0')}
UNLOAD-TRUCK {'?obj': 'p0', '?truck': 't0', '?loc': 'l00'} {('at', 'p0', 'l00')}
DRIVE-TRUCK {'?truck': 't0', '?loc-from': 'l00', '?loc-to': 'l00', '?city': 'c0'} {('at', 't0', 'l00')}
NoOp {} {('TRUCK', 't1')}
FLY-AIRPLANE {'?airplane': 'a0', '?loc-from': 'l10', '?loc-to': 'l10'} {('at', 'a0', 'l10')}
NoOp {} {('OBJ', 'p0')}
NoOp {} {('LOCATION', 'l11')}
DRIVE-TRUCK {'?truck': 't1', '?loc-from': 'l11', '?loc-to': 'l10', '?city': 'c1'} {('at', 't1', 'l10')}
NoOp {} {('CITY', 'c0')}
NoOp {} {('CITY', 'c1')}
DRIVE-TRUCK {'?truck': 't1', '?loc-from': 'l10', '?loc-to': 'l11', '?city': 'c1'} {('at', 't1', 'l11')}
**FLY-AIRPLANE {'?airplane': 'a0', '?loc-from': 'l00', '?loc-to': 'l10'} {('at', 'a0', 'l10')}**
DRIVE-TRUCK {'?truck': 't1', '?loc-from': 'l10', '?loc-to': 'l10', '?city': 'c1'} {('at', 't1', 'l10')}
NoOp {} {('at', 't0', 'l01')}

516

518

```
LOAD-AIRPLANE {'?obj': 'p0', '?airplane': 'a0', '?loc': 'l00'} {('in', 'p0',
'a0')}
NoOp {} {('LOCATION', 'l01')}
DRIVE-TRUCK {'?truck': 't0', '?loc-from': 'l01', '?loc-to': 'l00', '?city': 'c0'}
{('at', 't0', 'l00')}
NoOp {} {('AIRPLANE', 'a0')}
NoOp {} {('at', 't1', 'l10')}
NoOp {} {('in-city', 'l10', 'c1')}
NoOp {} {('at', 't1', 'l11')}
NoOp {} {('at', 'p0', 'l01')}
NoOp {} {('in-city', 'l00', 'c0')}
NoOp {} {('at', 't0', 'l00')}
NoOp {} {('LOCATION', 'l00')}
NoOp {} {('AIRPORT', 'l10')}
NoOp {} {('at', 'p0', 'l00')}
NoOp {} {('AIRPORT', 'l00')}
NoOp {} {('in', 'p0', 'a0')}
NoOp {} {('TRUCK', 't0')}
NoOp {} {('in', 'p0', 't0')}
NoOp {} {('at', 'a0', 'l00')}
NoOp {} {('in-city', 'l11', 'c1')}
NoOp {} {('at', 'a0', 'l10')}
NoOp {} {('LOCATION', 'l10')}
NoOp {} {('OBJ', 'p1')}
NoOp {} {('in-city', 'l01', 'c0')}
UNLOAD-TRUCK {'?obj': 'p0', '?truck': 't0', '?loc': 'l00'} {('at', 'p0', 'l00')}
NoOp {} {('TRUCK', 't1')}
NoOp {} {('OBJ', 'p0')}
NoOp {} {('LOCATION', 'l11')}
NoOp {} {('CITY', 'c0')}
NoOp {} {('CITY', 'c1')}
DRIVE-TRUCK {'?truck': 't1', '?loc-from': 'l10', '?loc-to': 'l11', '?city': 'c1'}
{('at', 't1', 'l11')}
FLY-AIRPLANE {'?airplane': 'a0', '?loc-from': 'l00', '?loc-to': 'l10'} {('at',
'a0', 'l10')}
NoOp {} {('at', 't0', 'l01')}
```

LOAD-AIRPLANE {'?obj': 'p0', '?airplane': 'a0', '?loc': 'l00'} {('in', 'p0', 'a0')}

**FLY-AIRPLANE {'?airplane': 'a0', '?loc-from': 'l00', '?loc-to': 'l10'} {('at', 'a0', 'l10')}**

DRIVE-TRUCK {'?truck': 't0', '?loc-from': 'l01', '?loc-to': 'l00', '?city': 'c0'} {('at', 't0', 'l00')}

DRIVE-TRUCK {'?truck': 't1', '?loc-from': 'l10', '?loc-to': 'l11', '?city': 'c1'} {('at', 't1', 'l11')}

UNLOAD-TRUCK {'?obj': 'p0', '?truck': 't0', '?loc': 'l00'} {('at', 'p0', 'l00')}

NoOp {} {('LOCATION', 'l01')}

NoOp {} {('AIRPLANE', 'a0')}

NoOp {} {('at', 't1', 'l10')}

NoOp {} {('in-city', 'l10', 'c1')}

NoOp {} {('at', 't1', 'l11')}

NoOp {} {('at', 'p0', 'l01')}

NoOp {} {('in-city', 'l00', 'c0')}

NoOp {} {('at', 't0', 'l00')}

NoOp {} {('LOCATION', 'l00')}

NoOp {} {('AIRPORT', 'l10')}

NoOp {} {('at', 'p0', 'l00')}

NoOp {} {('AIRPORT', 'l00')}

NoOp {} {('in', 'p0', 'a0')}

NoOp {} {('TRUCK', 't0')}

NoOp {} {('in', 'p0', 't0')}

NoOp {} {('at', 'a0', 'l00')}

NoOp {} {('in-city', 'l11', 'c1')}

NoOp {} {('at', 'a0', 'l10')}

NoOp {} {('LOCATION', 'l10')}

NoOp {} {('OBJ', 'p1')}

NoOp {} {('in-city', 'l01', 'c0')}

NoOp {} {('TRUCK', 't1')}

NoOp {} {('OBJ', 'p0')}

NoOp {} {('LOCATION', 'l11')}

NoOp {} {('CITY', 'c0')}

NoOp {} {('CITY', 'c1')}

NoOp {} {('at', 't0', 'l01')}

521     7.  Layer 7
522         GP-ACTION-SET

DRIVE-TRUCK {'?truck': 't1', '?loc-from': 'l11', '?loc-to': 'l10', '?city': 'c1'} {('at', 't1', 'l10')}
FLY-AIRPLANE {'?airplane': 'a0', '?loc-from': 'l10', '?loc-to': 'l10'} {('at', 'a0', 'l10')}
NoOp {} {('CITY', 'c0')}
DRIVE-TRUCK {'?truck': 't0', '?loc-from': 'l00', '?loc-to': 'l00', '?city': 'c0'} {('at', 't0', 'l00')}
NoOp {} {('in-city', 'l01', 'c0')}
NoOp {} {('AIRPLANE', 'a0')}
DRIVE-TRUCK {'?truck': 't0', '?loc-from': 'l00', '?loc-to': 'l01', '?city': 'c0'} {('at', 't0', 'l01')}
NoOp {} {('LOCATION', 'l00')}
DRIVE-TRUCK {'?truck': 't0', '?loc-from': 'l01', '?loc-to': 'l01', '?city': 'c0'} {('at', 't0', 'l01')}
UNLOAD-TRUCK {'?obj': 'p0', '?truck': 't0', '?loc': 'l01'} {('at', 'p0', 'l01')}
FLY-AIRPLANE {'?airplane': 'a0', '?loc-from': 'l00', '?loc-to': 'l10'} {('at', 'a0', 'l10')}
NoOp {} {('in', 'p0', 't0')}
NoOp {} {('in-city', 'l10', 'c1')}
NoOp {} {('LOCATION', 'l01')}
DRIVE-TRUCK {'?truck': 't1', '?loc-from': 'l10', '?loc-to': 'l11', '?city': 'c1'} {('at', 't1', 'l11')}
NoOp {} {('AIRPORT', 'l00')}
NoOp {} {('LOCATION', 'l11')}
NoOp {} {('TRUCK', 't1')}
NoOp {} {('in', 'p0', 'a0')}
UNLOAD-TRUCK {'?obj': 'p0', '?truck': 't0', '?loc': 'l00'} {('at', 'p0', 'l00')}
NoOp {} {('LOCATION', 'l10')}
NoOp {} {('at', 'p0', 'l01')}
NoOp {} {('CITY', 'c1')}
LOAD-TRUCK {'?obj': 'p0', '?truck': 't0', '?loc': 'l00'} {('in', 'p0', 't0')}
NoOp {} {('at', 't0', 'l00')}
NoOp {} {('TRUCK', 't0')}
NoOp {} {('at', 't0', 'l01')}
NoOp {} {('at', 't1', 'l10')}
NoOp {} {('AIRPORT', 'l10')}
NoOp {} {('OBJ', 'p0')}
**UNLOAD-AIRPLANE {'?obj': 'p0', '?airplane': 'a0', '?loc': 'l10'} {('at', 'p0', 'l10')}**

```
NoOp {} {('at', 't1', 'l11')}
DRIVE-TRUCK {'?truck': 't0', '?loc-from': 'l01', '?loc-to': 'l00', '?city': 'c0'}
{('at', 't0', 'l00')}
UNLOAD-AIRPLANE {'?obj': 'p0', '?airplane': 'a0', '?loc': 'l00'} {('at', 'p0',
'l00')}
NoOp {} {('at', 'a0', 'l10')}
DRIVE-TRUCK {'?truck': 't1', '?loc-from': 'l10', '?loc-to': 'l10', '?city': 'c1'}
{('at', 't1', 'l10')}
NoOp {} {('at', 'p0', 'l00')}
NoOp {} {('at', 'a0', 'l00')}
NoOp {} {('OBJ', 'p1')}
DRIVE-TRUCK {'?truck': 't1', '?loc-from': 'l11', '?loc-to': 'l11', '?city': 'c1'}
{('at', 't1', 'l11')}
LOAD-TRUCK {'?obj': 'p0', '?truck': 't0', '?loc': 'l01'} {('in', 'p0', 't0')}
FLY-AIRPLANE {'?airplane': 'a0', '?loc-from': 'l10', '?loc-to': 'l00'} {('at',
'a0', 'l00')}
NoOp {} {('in-city', 'l11', 'c1')}
NoOp {} {('in-city', 'l00', 'c0')}
LOAD-AIRPLANE {'?obj': 'p0', '?airplane': 'a0', '?loc': 'l00'} {('in', 'p0',
'a0')}
FLY-AIRPLANE {'?airplane': 'a0', '?loc-from': 'l00', '?loc-to': 'l00'} {('at',
'a0', 'l00')}
```

524

525     LLM-EP-ACTION-SET

```
NoOp {} {('CITY', 'c0')}
NoOp {} {('in-city', 'l01', 'c0')}
NoOp {} {('AIRPLANE', 'a0')}
NoOp {} {('LOCATION', 'l00')}
FLY-AIRPLANE {'?airplane': 'a0', '?loc-from': 'l00', '?loc-to': 'l10'} {('at',
'a0', 'l10')}
NoOp {} {('in', 'p0', 't0')}
NoOp {} {('in-city', 'l10', 'c1')}
NoOp {} {('LOCATION', 'l01')}
DRIVE-TRUCK {'?truck': 't1', '?loc-from': 'l10', '?loc-to': 'l11', '?city': 'c1'}
{('at', 't1', 'l11')}
NoOp {} {('AIRPORT', 'l00')}
NoOp {} {('LOCATION', 'l11')}
NoOp {} {('TRUCK', 't1')}
NoOp {} {('in', 'p0', 'a0')}
UNLOAD-TRUCK {'?obj': 'p0', '?truck': 't0', '?loc': 'l00'} {('at', 'p0', 'l00')}
NoOp {} {('LOCATION', 'l10')}
NoOp {} {('at', 'p0', 'l01')}
NoOp {} {('CITY', 'c1')}
NoOp {} {('at', 't0', 'l00')}
NoOp {} {('TRUCK', 't0')}
NoOp {} {('at', 't0', 'l01')}
NoOp {} {('at', 't1', 'l10')}
NoOp {} {('AIRPORT', 'l10')}
NoOp {} {('OBJ', 'p0')}
UNLOAD-AIRPLANE {'?obj': 'p0', '?airplane': 'a0', '?loc': 'l10'} {('at',
'p0', 'l10')}
NoOp {} {('at', 't1', 'l11')}
DRIVE-TRUCK {'?truck': 't0', '?loc-from': 'l01', '?loc-to': 'l00', '?city': 'c0'}
{('at', 't0', 'l00')}
NoOp {} {('at', 'a0', 'l10')}
NoOp {} {('at', 'p0', 'l00')}
NoOp {} {('at', 'a0', 'l00')}
NoOp {} {('OBJ', 'p1')}
NoOp {} {('in-city', 'l11', 'c1')}
NoOp {} {('in-city', 'l00', 'c0')}
LOAD-AIRPLANE {'?obj': 'p0', '?airplane': 'a0', '?loc': 'l00'} {('in', 'p0',
'a0')}
```

527    LLM-BP-ACTION-SET

**UNLOAD-AIRPLANE {'?obj': 'p0', '?airplane': 'a0', '?loc': 'l10'} {('at', 'p0', 'l10')}**
FLY-AIRPLANE {'?airplane': 'a0', '?loc-from': 'l00', '?loc-to': 'l10'} {('at', 'a0', 'l10')}
LOAD-AIRPLANE {'?obj': 'p0', '?airplane': 'a0', '?loc': 'l00'} {('in', 'p0', 'a0')}
DRIVE-TRUCK {'?truck': 't1', '?loc-from': 'l10', '?loc-to': 'l11', '?city': 'c1'} {('at', 't1', 'l11')}
UNLOAD-TRUCK {'?obj': 'p0', '?truck': 't0', '?loc': 'l00'} {('at', 'p0', 'l00')}
DRIVE-TRUCK {'?truck': 't0', '?loc-from': 'l01', '?loc-to': 'l00', '?city': 'c0'} {('at', 't0', 'l00')}
NoOp {} {('CITY', 'c0')}
NoOp {} {('in-city', 'l01', 'c0')}
NoOp {} {('AIRPLANE', 'a0')}
NoOp {} {('LOCATION', 'l00')}
NoOp {} {('in', 'p0', 't0')}
NoOp {} {('in-city', 'l10', 'c1')}
NoOp {} {('LOCATION', 'l01')}
NoOp {} {('AIRPORT', 'l00')}
NoOp {} {('LOCATION', 'l11')}
NoOp {} {('TRUCK', 't1')}
NoOp {} {('in', 'p0', 'a0')}
NoOp {} {('LOCATION', 'l10')}
NoOp {} {('at', 'p0', 'l01')}
NoOp {} {('CITY', 'c1')}
NoOp {} {('at', 't0', 'l00')}
NoOp {} {('TRUCK', 't0')}
NoOp {} {('at', 't0', 'l01')}
NoOp {} {('at', 't1', 'l10')}
NoOp {} {('AIRPORT', 'l10')}
NoOp {} {('OBJ', 'p0')}
NoOp {} {('at', 't1', 'l11')}
NoOp {} {('at', 'a0', 'l10')}
NoOp {} {('at', 'p0', 'l00')}
NoOp {} {('at', 'a0', 'l00')}
NoOp {} {('OBJ', 'p1')}
NoOp {} {('in-city', 'l11', 'c1')}
NoOp {} {('in-city', 'l00', 'c0')}

529      8. Layer 8
530          GP-ACTION-SET

UNLOAD-TRUCK {'?obj': 'p0', '?truck': 't0', '?loc': 'l00'} {('at', 'p0', 'l00')}
NoOp {} {('in', 'p0', 'a0')}
NoOp {} {('CITY', 'c0')}
DRIVE-TRUCK {'?truck': 't0', '?loc-from': 'l01', '?loc-to': 'l00', '?city': 'c0'}
{('at', 't0', 'l00')}
NoOp {} {('in-city', 'l01', 'c0')}
UNLOAD-AIRPLANE {'?obj': 'p0', '?airplane': 'a0', '?loc': 'l00'} {('at', 'p0',
'l00')}
LOAD-TRUCK {'?obj': 'p0', '?truck': 't0', '?loc': 'l00'} {('in', 'p0', 't0')}
NoOp {} {('AIRPLANE', 'a0')}
NoOp {} {('at', 't1', 'l10')}
UNLOAD-TRUCK {'?obj': 'p0', '?truck': 't0', '?loc': 'l01'} {('at', 'p0', 'l01')}
NoOp {} {('at', 'p0', 'l01')}
LOAD-TRUCK {'?obj': 'p0', '?truck': 't0', '?loc': 'l01'} {('in', 'p0', 't0')}
NoOp {} {('in-city', 'l00', 'c0')}
NoOp {} {('TRUCK', 't0')}
NoOp {} {('at', 't0', 'l00')}
NoOp {} {('OBJ', 'p1')}
DRIVE-TRUCK {'?truck': 't0', '?loc-from': 'l00', '?loc-to': 'l01', '?city': 'c0'}
{('at', 't0', 'l01')}
NoOp {} {('LOCATION', 'l00')}
NoOp {} {('at', 'p0', 'l00')}
FLY-AIRPLANE {'?airplane': 'a0', '?loc-from': 'l00', '?loc-to': 'l00'} {('at',
'a0', 'l00')}
LOAD-AIRPLANE {'?obj': 'p0', '?airplane': 'a0', '?loc': 'l00'} {('in', 'p0',
'a0')}
NoOp {} {('LOCATION', 'l01')}
NoOp {} {('LOCATION', 'l10')}
**LOAD-TRUCK {'?obj': 'p0', '?truck': 't1', '?loc': 'l10'} {('in', 'p0', 't1')}**

```
NoOp {} {('at', 't1', 'l11')}
DRIVE-TRUCK {'?truck': 't0', '?loc-from': 'l01', '?loc-to': 'l01', '?city': 'c0'}
{('at', 't0', 'l01')}
DRIVE-TRUCK {'?truck': 't1', '?loc-from': 'l11', '?loc-to': 'l10', '?city': 'c1'}
{('at', 't1', 'l10')}
FLY-AIRPLANE {'?airplane': 'a0', '?loc-from': 'l10', '?loc-to': 'l00'} {('at',
'a0', 'l00')}
NoOp {} {('at', 'p0', 'l10')}
NoOp {} {('at', 'a0', 'l00')}
FLY-AIRPLANE {'?airplane': 'a0', '?loc-from': 'l00', '?loc-to': 'l10'} {('at',
'a0', 'l10')}
UNLOAD-AIRPLANE {'?obj': 'p0', '?airplane': 'a0', '?loc': 'l10'} {('at', 'p0',
'l10')}
NoOp {} {('CITY', 'c1')}
FLY-AIRPLANE {'?airplane': 'a0', '?loc-from': 'l10', '?loc-to': 'l10'} {('at',
'a0', 'l10')}
NoOp {} {('AIRPORT', 'l10')}
NoOp {} {('in', 'p0', 't0')}
DRIVE-TRUCK {'?truck': 't0', '?loc-from': 'l00', '?loc-to': 'l00', '?city': 'c0'}
{('at', 't0', 'l00')}
DRIVE-TRUCK {'?truck': 't1', '?loc-from': 'l10', '?loc-to': 'l10', '?city': 'c1'}
{('at', 't1', 'l10')}
NoOp {} {('OBJ', 'p0')}
NoOp {} {('at', 'a0', 'l10')}
DRIVE-TRUCK {'?truck': 't1', '?loc-from': 'l10', '?loc-to': 'l11', '?city': 'c1'}
{('at', 't1', 'l11')}
NoOp {} {('in-city', 'l10', 'c1')}
LOAD-AIRPLANE {'?obj': 'p0', '?airplane': 'a0', '?loc': 'l10'} {('in', 'p0',
'a0')}
NoOp {} {('at', 't0', 'l01')}
DRIVE-TRUCK {'?truck': 't1', '?loc-from': 'l11', '?loc-to': 'l11', '?city': 'c1'}
{('at', 't1', 'l11')}
NoOp {} {('in-city', 'l11', 'c1')}
NoOp {} {('AIRPORT', 'l00')}
NoOp {} {('LOCATION', 'l11')}
NoOp {} {('TRUCK', 't1')}
```

LLM-EP-ACTION-SET

```
UNLOAD-TRUCK {'?obj': 'p0', '?truck': 't0', '?loc': 'l00'} {('at', 'p0', 'l00')}
NoOp {} {('in', 'p0', 'a0')}
NoOp {} {('CITY', 'c0')}
DRIVE-TRUCK {'?truck': 't0', '?loc-from': 'l01', '?loc-to': 'l00', '?city': 'c0'}
{('at', 't0', 'l00')}
NoOp {} {('in-city', 'l01', 'c0')}
NoOp {} {('AIRPLANE', 'a0')}
NoOp {} {('at', 't1', 'l10')}
NoOp {} {('at', 'p0', 'l01')}
NoOp {} {('in-city', 'l00', 'c0')}
NoOp {} {('TRUCK', 't0')}
NoOp {} {('at', 't0', 'l00')}
NoOp {} {('OBJ', 'p1')}
NoOp {} {('LOCATION', 'l00')}
NoOp {} {('at', 'p0', 'l00')}
LOAD-AIRPLANE {'?obj': 'p0', '?airplane': 'a0', '?loc': 'l00'} {('in', 'p0',
'a0')}
NoOp {} {('LOCATION', 'l01')}
NoOp {} {('LOCATION', 'l10')}
LOAD-TRUCK {'?obj': 'p0', '?truck': 't1', '?loc': 'l10'} {('in', 'p0', 't1')}
NoOp {} {('at', 't1', 'l11')}
NoOp {} {('at', 'p0', 'l10')}
NoOp {} {('at', 'a0', 'l00')}
FLY-AIRPLANE {'?airplane': 'a0', '?loc-from': 'l00', '?loc-to': 'l10'} {('at',
'a0', 'l10')}
UNLOAD-AIRPLANE {'?obj': 'p0', '?airplane': 'a0', '?loc': 'l10'} {('at', 'p0',
'l10')}
NoOp {} {('CITY', 'c1')}
NoOp {} {('AIRPORT', 'l10')}
NoOp {} {('in', 'p0', 't0')}
NoOp {} {('OBJ', 'p0')}
NoOp {} {('at', 'a0', 'l10')}
DRIVE-TRUCK {'?truck': 't1', '?loc-from': 'l10', '?loc-to': 'l11', '?city': 'c1'}
{('at', 't1', 'l11')}
NoOp {} {('in-city', 'l10', 'c1')}
NoOp {} {('at', 't0', 'l01')}
NoOp {} {('in-city', 'l11', 'c1')}
NoOp {} {('AIRPORT', 'l00')}
NoOp {} {('LOCATION', 'l11')}
NoOp {} {('TRUCK', 't1')}
```

LLM-BP-ACTION-SET

DRIVE-TRUCK {'?truck': 't1', '?loc-from': 'l10', '?loc-to': 'l11', '?city': 'c1'} {('at', 't1', 'l11')}

**LOAD-TRUCK {'?obj': 'p0', '?truck': 't1', '?loc': 'l10'} {('in', 'p0', 't1')}**

UNLOAD-TRUCK {'?obj': 'p0', '?truck': 't0', '?loc': 'l00'} {('at', 'p0', 'l00')}

DRIVE-TRUCK {'?truck': 't0', '?loc-from': 'l01', '?loc-to': 'l00', '?city': 'c0'} {('at', 't0', 'l00')}

LOAD-AIRPLANE {'?obj': 'p0', '?airplane': 'a0', '?loc': 'l00'} {('in', 'p0', 'a0')}

FLY-AIRPLANE {'?airplane': 'a0', '?loc-from': 'l00', '?loc-to': 'l10'} {('at', 'a0', 'l10')}

UNLOAD-AIRPLANE {'?obj': 'p0', '?airplane': 'a0', '?loc': 'l10'} {('at', 'p0', 'l10')}

NoOp {} {('in', 'p0', 'a0')}
NoOp {} {('CITY', 'c0')}
NoOp {} {('in-city', 'l01', 'c0')}
NoOp {} {('AIRPLANE', 'a0')}
NoOp {} {('at', 't1', 'l10')}
NoOp {} {('at', 'p0', 'l01')}
NoOp {} {('in-city', 'l00', 'c0')}
NoOp {} {('TRUCK', 't0')}
NoOp {} {('at', 't0', 'l00')}
NoOp {} {('OBJ', 'p1')}
NoOp {} {('LOCATION', 'l00')}
NoOp {} {('at', 'p0', 'l00')}
NoOp {} {('LOCATION', 'l01')}
NoOp {} {('LOCATION', 'l10')}
NoOp {} {('at', 't1', 'l11')}
NoOp {} {('at', 'p0', 'l10')}
NoOp {} {('at', 'a0', 'l00')}
NoOp {} {('CITY', 'c1')}
NoOp {} {('AIRPORT', 'l10')}
NoOp {} {('in', 'p0', 't0')}
NoOp {} {('OBJ', 'p0')}
NoOp {} {('at', 'a0', 'l10')}
NoOp {} {('in-city', 'l10', 'c1')}
NoOp {} {('at', 't0', 'l01')}
NoOp {} {('in-city', 'l11', 'c1')}
NoOp {} {('AIRPORT', 'l00')}
NoOp {} {('LOCATION', 'l11')}
NoOp {} {('TRUCK', 't1')}

9. Layer 9

    GP-ACTION-SET

LOAD-TRUCK {'?obj': 'p0', '?truck': 't0', '?loc': 'l00'} {('in', 'p0', 't0')}
NoOp {} {('at', 'p0', 'l00')}
LOAD-TRUCK {'?obj': 'p0', '?truck': 't0', '?loc': 'l01'} {('in', 'p0', 't0')}
UNLOAD-AIRPLANE {'?obj': 'p0', '?airplane': 'a0', '?loc': 'l00'} {('at', 'p0', 'l00')}
DRIVE-TRUCK {'?truck': 't0', '?loc-from': 'l00', '?loc-to': 'l00', '?city': 'c0'} {('at', 't0', 'l00')}
DRIVE-TRUCK {'?truck': 't0', '?loc-from': 'l01', '?loc-to': 'l01', '?city': 'c0'} {('at', 't0', 'l01')}
NoOp {} {('in', 'p0', 't0')}
UNLOAD-AIRPLANE {'?obj': 'p0', '?airplane': 'a0', '?loc': 'l10'} {('at', 'p0', 'l10')}
DRIVE-TRUCK {'?truck': 't0', '?loc-from': 'l00', '?loc-to': 'l01', '?city': 'c0'} {('at', 't0', 'l01')}
NoOp {} {('at', 'p0', 'l01')}
NoOp {} {('OBJ', 'p1')}
NoOp {} {('in-city', 'l11', 'c1')}
NoOp {} {('at', 't0', 'l00')}
NoOp {} {('AIRPLANE', 'a0')}
NoOp {} {('in-city', 'l10', 'c1')}
DRIVE-TRUCK {'?truck': 't0', '?loc-from': 'l01', '?loc-to': 'l00', '?city': 'c0'} {('at', 't0', 'l00')}
UNLOAD-TRUCK {'?obj': 'p0', '?truck': 't0', '?loc': 'l01'} {('at', 'p0', 'l01')}
NoOp {} {('LOCATION', 'l11')}
NoOp {} {('AIRPORT', 'l10')}
UNLOAD-TRUCK {'?obj': 'p0', '?truck': 't1', '?loc': 'l10'} {('at', 'p0', 'l10')}
**DRIVE-TRUCK {'?truck': 't1', '?loc-from': 'l10', '?loc-to': 'l11', '?city': 'c1'} {('at', 't1', 'l11')}**

```
NoOp {} {('TRUCK', 't0')}
NoOp {} {('TRUCK', 't1')}
NoOp {} {('at', 'p0', 'l10')}
FLY-AIRPLANE {'?airplane': 'a0', '?loc-from': 'l00', '?loc-to': 'l10'} {('at',
'a0', 'l10')}
NoOp {} {('at', 't1', 'l11')}
NoOp {} {('in-city', 'l01', 'c0')}
LOAD-AIRPLANE {'?obj': 'p0', '?airplane': 'a0', '?loc': 'l00'} {('in', 'p0',
'a0')}
DRIVE-TRUCK {'?truck': 't1', '?loc-from': 'l10', '?loc-to': 'l10', '?city': 'c1'}
{('at', 't1', 'l10')}
NoOp {} {('at', 't1', 'l10')}
NoOp {} {('in', 'p0', 't1')}
NoOp {} {('in-city', 'l00', 'c0')}
NoOp {} {('CITY', 'c1')}
NoOp {} {('LOCATION', 'l10')}
NoOp {} {('in', 'p0', 'a0')}
NoOp {} {('OBJ', 'p0')}
LOAD-TRUCK {'?obj': 'p0', '?truck': 't1', '?loc': 'l10'} {('in', 'p0', 't1')}
NoOp {} {('LOCATION', 'l01')}
FLY-AIRPLANE {'?airplane': 'a0', '?loc-from': 'l00', '?loc-to': 'l00'} {('at',
'a0', 'l00')}
NoOp {} {('LOCATION', 'l00')}
FLY-AIRPLANE {'?airplane': 'a0', '?loc-from': 'l10', '?loc-to': 'l10'} {('at',
'a0', 'l10')}
DRIVE-TRUCK {'?truck': 't1', '?loc-from': 'l11', '?loc-to': 'l11', '?city': 'c1'}
{('at', 't1', 'l11')}
UNLOAD-TRUCK {'?obj': 'p0', '?truck': 't0', '?loc': 'l00'} {('at', 'p0', 'l00')}
DRIVE-TRUCK {'?truck': 't1', '?loc-from': 'l11', '?loc-to': 'l10', '?city': 'c1'}
{('at', 't1', 'l10')}
NoOp {} {('AIRPORT', 'l00')}
NoOp {} {('at', 't0', 'l01')}
FLY-AIRPLANE {'?airplane': 'a0', '?loc-from': 'l10', '?loc-to': 'l00'} {('at',
'a0', 'l00')}
NoOp {} {('CITY', 'c0')}
LOAD-AIRPLANE {'?obj': 'p0', '?airplane': 'a0', '?loc': 'l10'} {('in', 'p0',
'a0')}
NoOp {} {('at', 'a0', 'l00')}
NoOp {} {('at', 'a0', 'l10')}
```

540

541       LLM-EP-ACTION-SET

NoOp {} {('at', 'p0', 'l00')}
NoOp {} {('in', 'p0', 't0')}
UNLOAD-AIRPLANE {'?obj': 'p0', '?airplane': 'a0', '?loc': 'l10'} {('at', 'p0', 'l10')}
NoOp {} {('at', 'p0', 'l01')}
NoOp {} {('OBJ', 'p1')}
NoOp {} {('in-city', 'l11', 'c1')}
NoOp {} {('at', 't0', 'l00')}
NoOp {} {('AIRPLANE', 'a0')}
NoOp {} {('in-city', 'l10', 'c1')}
DRIVE-TRUCK {'?truck': 't0', '?loc-from': 'l01', '?loc-to': 'l00', '?city': 'c0'} {('at', 't0', 'l00')}
NoOp {} {('LOCATION', 'l11')}
NoOp {} {('AIRPORT', 'l10')}
**DRIVE-TRUCK {'?truck': 't1', '?loc-from': 'l10', '?loc-to': 'l11', '?city': 'c1'} {('at', 't1', 'l11')}**
NoOp {} {('TRUCK', 't0')}
NoOp {} {('TRUCK', 't1')}
NoOp {} {('at', 'p0', 'l10')}
FLY-AIRPLANE {'?airplane': 'a0', '?loc-from': 'l00', '?loc-to': 'l10'} {('at', 'a0', 'l10')}
NoOp {} {('at', 't1', 'l11')}
NoOp {} {('in-city', 'l01', 'c0')}
LOAD-AIRPLANE {'?obj': 'p0', '?airplane': 'a0', '?loc': 'l00'} {('in', 'p0', 'a0')}
NoOp {} {('at', 't1', 'l10')}
NoOp {} {('in', 'p0', 't1')}
NoOp {} {('in-city', 'l00', 'c0')}
NoOp {} {('CITY', 'c1')}
NoOp {} {('LOCATION', 'l10')}
NoOp {} {('in', 'p0', 'a0')}
NoOp {} {('OBJ', 'p0')}
NoOp {} {('LOCATION', 'l01')}
NoOp {} {('LOCATION', 'l00')}
UNLOAD-TRUCK {'?obj': 'p0', '?truck': 't0', '?loc': 'l00'} {('at', 'p0', 'l00')}
NoOp {} {('AIRPORT', 'l00')}
NoOp {} {('at', 't0', 'l01')}
NoOp {} {('CITY', 'c0')}
NoOp {} {('at', 'a0', 'l00')}
NoOp {} {('at', 'a0', 'l10')}

LLM-BP-ACTION-SET

**DRIVE-TRUCK {'?truck': 't1', '?loc-from': 'l10', '?loc-to': 'l11', '?city': 'c1'} {('at', 't1', 'l11')}**
UNLOAD-AIRPLANE {'?obj': 'p0', '?airplane': 'a0', '?loc': 'l10'} {('at', 'p0', 'l10')}
DRIVE-TRUCK {'?truck': 't0', '?loc-from': 'l01', '?loc-to': 'l00', '?city': 'c0'} {('at', 't0', 'l00')}
FLY-AIRPLANE {'?airplane': 'a0', '?loc-from': 'l00', '?loc-to': 'l10'} {('at', 'a0', 'l10')}
LOAD-AIRPLANE {'?obj': 'p0', '?airplane': 'a0', '?loc': 'l00'} {('in', 'p0', 'a0')}
UNLOAD-TRUCK {'?obj': 'p0', '?truck': 't0', '?loc': 'l00'} {('at', 'p0', 'l00')}
NoOp {} {('at', 'p0', 'l00')}
NoOp {} {('in', 'p0', 't0')}
NoOp {} {('at', 'p0', 'l01')}
NoOp {} {('OBJ', 'p1')}
NoOp {} {('in-city', 'l11', 'c1')}
NoOp {} {('at', 't0', 'l00')}
NoOp {} {('AIRPLANE', 'a0')}
NoOp {} {('in-city', 'l10', 'c1')}
NoOp {} {('LOCATION', 'l11')}
NoOp {} {('AIRPORT', 'l10')}
NoOp {} {('TRUCK', 't0')}
NoOp {} {('TRUCK', 't1')}
NoOp {} {('at', 'p0', 'l10')}
NoOp {} {('at', 't1', 'l11')}
NoOp {} {('in-city', 'l01', 'c0')}
NoOp {} {('at', 't1', 'l10')}
NoOp {} {('in', 'p0', 't1')}
NoOp {} {('in-city', 'l00', 'c0')}
NoOp {} {('CITY', 'c1')}
NoOp {} {('LOCATION', 'l10')}
NoOp {} {('in', 'p0', 'a0')}
NoOp {} {('OBJ', 'p0')}
NoOp {} {('LOCATION', 'l01')}
NoOp {} {('LOCATION', 'l00')}
NoOp {} {('AIRPORT', 'l00')}
NoOp {} {('at', 't0', 'l01')}
NoOp {} {('CITY', 'c0')}
NoOp {} {('at', 'a0', 'l00')}
NoOp {} {('at', 'a0', 'l10')}

10. Layer 10
    GP-ACTION-SET

NoOp {} {('at', 'a0', 'l00')}
NoOp {} {('at', 't1', 'l10')}
DRIVE-TRUCK {'?truck': 't0', '?loc-from': 'l00', '?loc-to': 'l01', '?city': 'c0'} {('at', 't0', 'l01')}
DRIVE-TRUCK {'?truck': 't1', '?loc-from': 'l11', '?loc-to': 'l11', '?city': 'c1'} {('at', 't1', 'l11')}
NoOp {} {('OBJ', 'p0')}
LOAD-AIRPLANE {'?obj': 'p0', '?airplane': 'a0', '?loc': 'l00'} {('in', 'p0', 'a0')}
NoOp {} {('in', 'p0', 'a0')}
DRIVE-TRUCK {'?truck': 't1', '?loc-from': 'l10', '?loc-to': 'l11', '?city': 'c1'} {('at', 't1', 'l11')}
NoOp {} {('AIRPORT', 'l00')}
DRIVE-TRUCK {'?truck': 't0', '?loc-from': 'l01', '?loc-to': 'l00', '?city': 'c0'} {('at', 't0', 'l00')}
NoOp {} {('in-city', 'l11', 'c1')}
LOAD-TRUCK {'?obj': 'p0', '?truck': 't0', '?loc': 'l01'} {('in', 'p0', 't0')}
NoOp {} {('at', 't1', 'l11')}
NoOp {} {('at', 'a0', 'l10')}
**UNLOAD-TRUCK {'?obj': 'p0', '?truck': 't1', '?loc': 'l11'} {('at', 'p0', 'l11')}**

547

NoOp {} {('at', 'p0', 'l01')}
NoOp {} {('in-city', 'l10', 'c1')}
DRIVE-TRUCK {'?truck': 't0', '?loc-from': 'l00', '?loc-to': 'l00', '?city': 'c0'} {('at', 't0', 'l00')}
DRIVE-TRUCK {'?truck': 't1', '?loc-from': 'l11', '?loc-to': 'l10', '?city': 'c1'} {('at', 't1', 'l10')}
FLY-AIRPLANE {'?airplane': 'a0', '?loc-from': 'l00', '?loc-to': 'l00'} {('at', 'a0', 'l00')}
FLY-AIRPLANE {'?airplane': 'a0', '?loc-from': 'l00', '?loc-to': 'l10'} {('at', 'a0', 'l10')}
UNLOAD-AIRPLANE {'?obj': 'p0', '?airplane': 'a0', '?loc': 'l10'} {('at', 'p0', 'l10')}
NoOp {} {('in-city', 'l00', 'c0')}
NoOp {} {('AIRPORT', 'l10')}
NoOp {} {('AIRPLANE', 'a0')}
NoOp {} {('in', 'p0', 't1')}
NoOp {} {('at', 'p0', 'l10')}
NoOp {} {('OBJ', 'p1')}
NoOp {} {('LOCATION', 'l10')}
UNLOAD-TRUCK {'?obj': 'p0', '?truck': 't0', '?loc': 'l00'} {('at', 'p0', 'l00')}
NoOp {} {('at', 'p0', 'l00')}
UNLOAD-TRUCK {'?obj': 'p0', '?truck': 't0', '?loc': 'l01'} {('at', 'p0', 'l01')}
NoOp {} {('CITY', 'c1')}
LOAD-TRUCK {'?obj': 'p0', '?truck': 't0', '?loc': 'l00'} {('in', 'p0', 't0')}
NoOp {} {('TRUCK', 't0')}
UNLOAD-TRUCK {'?obj': 'p0', '?truck': 't1', '?loc': 'l10'} {('at', 'p0', 'l10')}
NoOp {} {('at', 't0', 'l01')}
NoOp {} {('LOCATION', 'l11')}
FLY-AIRPLANE {'?airplane': 'a0', '?loc-from': 'l10', '?loc-to': 'l10'} {('at', 'a0', 'l10')}
NoOp {} {('at', 't0', 'l00')}
FLY-AIRPLANE {'?airplane': 'a0', '?loc-from': 'l10', '?loc-to': 'l00'} {('at', 'a0', 'l00')}
DRIVE-TRUCK {'?truck': 't1', '?loc-from': 'l10', '?loc-to': 'l10', '?city': 'c1'} {('at', 't1', 'l10')}
NoOp {} {('LOCATION', 'l01')}
DRIVE-TRUCK {'?truck': 't0', '?loc-from': 'l01', '?loc-to': 'l01', '?city': 'c0'} {('at', 't0', 'l01')}
NoOp {} {('CITY', 'c0')}
NoOp {} {('LOCATION', 'l00')}
UNLOAD-AIRPLANE {'?obj': 'p0', '?airplane': 'a0', '?loc': 'l00'} {('at', 'p0', 'l00')}
NoOp {} {('in-city', 'l01', 'c0')}
NoOp {} {('in', 'p0', 't0')}
LOAD-AIRPLANE {'?obj': 'p0', '?airplane': 'a0', '?loc': 'l10'} {('in', 'p0', 'a0')}
NoOp {} {('TRUCK', 't1')}
LOAD-TRUCK {'?obj': 'p0', '?truck': 't1', '?loc': 'l10'} {('in', 'p0', 't1')}

549        LLM-EP-ACTION-SET

```
NoOp {} {('at', 'a0', 'l00')}
NoOp {} {('at', 't1', 'l10')}
NoOp {} {('OBJ', 'p0')}
NoOp {} {('in', 'p0', 'a0')}
DRIVE-TRUCK {'?truck': 't1', '?loc-from': 'l10', '?loc-to': 'l11', '?city': 'c1'}
{('at', 't1', 'l11')}
NoOp {} {('AIRPORT', 'l00')}
NoOp {} {('in-city', 'l11', 'c1')}
NoOp {} {('at', 't1', 'l11')}
NoOp {} {('at', 'a0', 'l10')}
```
**UNLOAD-TRUCK {'?obj': 'p0', '?truck': 't1', '?loc': 'l11'} {('at', 'p0', 'l11')}**
```
NoOp {} {('at', 'p0', 'l01')}
NoOp {} {('in-city', 'l10', 'c1')}
NoOp {} {('in-city', 'l00', 'c0')}
NoOp {} {('AIRPORT', 'l10')}
NoOp {} {('AIRPLANE', 'a0')}
NoOp {} {('in', 'p0', 't1')}
NoOp {} {('at', 'p0', 'l10')}
NoOp {} {('OBJ', 'p1')}
NoOp {} {('LOCATION', 'l10')}
NoOp {} {('at', 'p0', 'l00')}
NoOp {} {('CITY', 'c1')}
NoOp {} {('TRUCK', 't0')}
NoOp {} {('at', 't0', 'l01')}
NoOp {} {('LOCATION', 'l11')}
NoOp {} {('at', 't0', 'l00')}
NoOp {} {('LOCATION', 'l01')}
NoOp {} {('CITY', 'c0')}
NoOp {} {('LOCATION', 'l00')}
NoOp {} {('in-city', 'l01', 'c0')}
NoOp {} {('in', 'p0', 't0')}
NoOp {} {('TRUCK', 't1')}
LOAD-TRUCK {'?obj': 'p0', '?truck': 't1', '?loc': 'l10'} {('in', 'p0', 't1')}
```

LLM-BP-ACTION-SET

DRIVE-TRUCK {'?truck': 't1', '?loc-from': 'l10', '?loc-to': 'l11', '?city': 'c1'} {('at', 't1', 'l11')}

**UNLOAD-TRUCK {'?obj': 'p0', '?truck': 't1', '?loc': 'l11'} {('at', 'p0', 'l11')}**

LOAD-TRUCK {'?obj': 'p0', '?truck': 't1', '?loc': 'l10'} {('in', 'p0', 't1')}
NoOp {} {('at', 'a0', 'l00')}
NoOp {} {('at', 't1', 'l10')}
NoOp {} {('OBJ', 'p0')}
NoOp {} {('in', 'p0', 'a0')}
NoOp {} {('AIRPORT', 'l00')}
NoOp {} {('in-city', 'l11', 'c1')}
NoOp {} {('at', 't1', 'l11')}
NoOp {} {('at', 'a0', 'l10')}
NoOp {} {('at', 'p0', 'l01')}
NoOp {} {('in-city', 'l10', 'c1')}
NoOp {} {('in-city', 'l00', 'c0')}
NoOp {} {('AIRPORT', 'l10')}
NoOp {} {('AIRPLANE', 'a0')}
NoOp {} {('in', 'p0', 't1')}
NoOp {} {('at', 'p0', 'l10')}
NoOp {} {('OBJ', 'p1')}
NoOp {} {('LOCATION', 'l10')}
NoOp {} {('at', 'p0', 'l00')}
NoOp {} {('CITY', 'c1')}
NoOp {} {('TRUCK', 't0')}
NoOp {} {('at', 't0', 'l01')}
NoOp {} {('LOCATION', 'l11')}
NoOp {} {('at', 't0', 'l00')}
NoOp {} {('LOCATION', 'l01')}
NoOp {} {('CITY', 'c0')}
NoOp {} {('LOCATION', 'l00')}
NoOp {} {('in-city', 'l01', 'c0')}
NoOp {} {('in', 'p0', 't0')}
NoOp {} {('TRUCK', 't1')}

