# OpenReview forum: "On the Roles of LLMs in Planning: Embedding LLMs into Planning Graphs"
_NeurIPS.cc/2024/Conference — Submitted to NeurIPS 2024_

### Official Review · Reviewer_KX5q · 2024-06-14

**Soundness:** 3
**Presentation:** 2
**Contribution:** 3
**Rating:** 6
**Confidence:** 4

**Summary:**

This work creates a hybrid LLM and classic planning algorithm, by integrating a LLM into the GraphPlan algorithm. The GraphPlan is an algorithm that solves a relaxed planning problem (forward expansion), and then traverses the created graph to find a valid plan (backtracking). Both steps are expensive. In the hybrid approach, a LLM is prompted in the forward expansion to limit the exploration of states deemed irrelevant. In the backtracking phase, the LLM is used to sort actions to explore first. Experiments with corrupted domain files show that LLMs can better handle corruption than the GraphPlan algorithm.

**Strengths:**

- A very interesting novel idea of a hybrid planning approach with a fundamental classic planning algorithm.
- The paper provides an introduction to an interesting research area of classical planning (e.g., Figure 4).

**Weaknesses:**

- Multiple missing experiments and discussions severely undermine the results of the paper.
    - It is not clearly motivated why experiments with corrupted pddl domain files are interesting. This was introduced quite suddenly in the *results section* (lines 261-262) without enough details and without providing motivation.
    - The paper is missing important discussion and experiments about the trade-off between the hybrid approach and the classic GP algorithm. Experiments with valid pddl domain files are not included, which could have alleviate it.
    - The effect of hyperparameters on the results, such as the number of iterations (N) in Algorithm 1, is not discussed.
    - The failure of LLMs4PLAN-GPT3.5 compared to the phenomenal success of LLMs4Plan-GPT4 is somewhat unexpected and undermines the results of the paper.
- Multiple details are missing regarding the experimental setups. (see questions below)
- The paper's writing needs to be improved. (see suggestions below)

**Questions:**

**Suggestions**
- Most importantly, I would like to see experiments with valid pddl files. While the GP algorithm would receive 100% success rate, I expected to see a graph of success rate (y-axis) compared to the number of nodes explored (x-axis). Such a graph would describe **a trade-off** between statistically using a LLM and using an exhaustive algorithm, such as GraphPlan.
    - It is possible that this is included in Table 3, but I don't understand if it includes the corrupted domain files. A graph which includes success rate would be clearer.
- Clearly describe and motivate the pddl data corruption in the experimental setup.
- Writing
    - While I happen to be well-versed with the GRAPHPLAN algorithm, I am not sure that enough introduction has been provided, as it is only briefly mentioned in lines 44 and 67.
    - “3 Our LLMs4Plan approach” does not properly introduce the algorithm before discussing it. Notations, such as N, are not defined. Terms, such as planning graphs and mutual constraints, are used but not explained until reading 3.1 and 3.2. Concepts, such as pruning, should be formally introduced before discussing “pruning possibility” (line 109).
    - subsection 3.2:
         - this subsection is not part of your algorithm, but part of graphplan. related to my previous notes, I think this should not be in section 3, but properly explained earlier.
         - add citations to support the mutual exclusion constraints names (i.e., inconsistent effects, interference, competing needs).
     - subsection 4.3. writing could be improved. I was initially confused about the location of the ablation results table.

**Questions**

- Data corruption of pddl domain files: Please provide details about the corruption, such as % files corrupted. It seems that this is a very high percentage, if this is the main reason that the GP method gets low resulsts (lines 261-262).
- Please provide details about the hyperparameters influence on the results, such as the number of iterations N and the number of layers K.
- In Table 2, where we compare “number of nodes required for searching”. Which iteration do we measure? Do we ignore the fact that there could have been multiple iterations before the successful one?
- line 189 - “Ten problems are randomly selected for each domain” - from which corpus?
- Statistics about the pddl problem files are missing. How long are the plans from initial state to goal?

**Limitations:**

The authors did not discuss limitations of their work.

---

> ### Author Rebuttal · Authors · 2024-08-07
>
> [Motivation of corrupted domain models]
> Response #4.1: In real-world applications it is often difficult to design complete domain models (without corruption) provided for classical planners to solve real-world planning problems [16]. It is an open challenging problem to design effective approach to learn domain models from history data. Designing an effective planning approach to solve problems with corrupted domain models would broaden its applications in real-world domains.
>
> [Experiments on valid domain models]
> Response #4.2: The numbers of expanded nodes on 5 valid questions of each domain (i.e., with valid domain models without corruption) are shown in the following table. As we can see from the table, the number of expanded nodes of our LLMs4Plan is much smaller than GP in valid domain models. We will add the corresponding results and analysis in the paper if the paper is accepted.
>
>  LLMs4Plan	LLMs4Plan-unsorted	LLMs4Plan-unpruned	GP
>
> gripper	8640	13240	6458928	11294380
>
> miconic	17818	71715	667647	3026119
>
> logistics	72	108	1628	1681658
>
> movie	1407182	1410930	1407182	14911545
>
> blocks	5510	9811	107334	1531608
>
> satellite	47315	104582	72573129	91379642
>
> zenotravel	1695	14428	892472	3067223
>
> driverlog	591	2281	70756	1842905
>
> woodworking	65	5216	74599	178890
>
> openstacks	111	478	18374	31292
>
>
> [Description of hyperparameters N and K]
> Response #4.3: N is set to be large enough to ensure the completeness of the approach. As we can see from Algorithm 1, when the iteration i (<N) becomes large enough, the pruning possibility \kapa_i is very small, making Algorithm 1 close to the classic GP algorithm. In our experiment, we empirically set N to be 7. K is the maximal number of layers to be expanded, which is related to the length of solution plans to planning problems. We empirically set K to be 25. Note that since there are parallel actions in each action layer, the length of final solution plans is generally much longer than the number of layers expanded. We will add the corresponding descriptions of hyperparameters in the paper if the paper is accepted.
>
> [The percentages of corruption]
> Response #4.4: Among 10 questions of each domain, 5 questions are without corruption, and other 5 questions are corrupted with 10%, 20%, 30%, 40%, 50% of preconditions and effects randomly removed in each action model.
>
> [How do we count the expanded nodes]
> Response #4.5: We counted all nodes expanded in multiple iterations, including repeated nodes expanded multiple times in multiple iterations.
>
> [How do we generate ten problems of each domain]
> Response #4.6:
> We used code libraries for generation of random problems (https://github.com/AI-Planning/pddl-generators/tree/main). The corresponding description is given in Appendix A.1. We will add more detailed descriptions about the generation procedure in Appendix A.1.
>
> [Length of solution plans]
> Response #4.7: The average of solution plans in each domain is between 15 and 80.

---

> > ### Comment · Reviewer_KX5q · 2024-08-08
> > **Response to rebuttal**
> >
> > Thank you to the authors for the new results. These align more closely with the expected experimental setup based on the introduction's motivation.
> >
> > The newly reported results are very impressive. Assuming these results and the additional details will be included in the paper, I am inclined to raise my score. However, my new score still considers the need for several modifications to improve the paper's clarity and the limitation of having only five non-corrupted problem files per domain.
> >
> > > when the iteration i (<N) becomes large enough, the pruning possibility \kapa_i is very small, making Algorithm 1 close to the classic GP algorithm
> >
> > I would consider adding another figure that depicts this trade-off between GP and your approach. x-axis is the number of iterations and y-axis is success rate. This will shed light on the number of iterations that were actually necessary.

---

> > > ### Author Response · Authors · 2024-08-08
> > >
> > > Thank you very much for your consideration and your further suggestion. We will add the new experimental results and revise the descriptions to improve the paper's clarity accordingly. Indeed, depicting the trade-off between GP and our proposed approach with respect to the number of iterations would give more insights on our approach. We will add the results to the paper accordingly (we collected the results before).

---

### Official Review · Reviewer_jjfw · 2024-06-27

**Soundness:** 3
**Presentation:** 2
**Contribution:** 3
**Rating:** 5
**Confidence:** 3

**Summary:**

The paper investigates how large language models (LLMs) can be integrated into established planning frameworks, specifically graph-based planning. The authors propose a novel framework called LLMs4Plan, which incorporates LLMs at two critical stages of the planning process: action selection during graph expansion and candidate action set generation during backtracking. The framework is tested across various planning domains, demonstrating improved efficiency and effectiveness in planning tasks.

**Strengths:**

1. The paper's approach of embedding LLMs into graph-based planning is innovative and contributes to the field of automated planning.
2. The technical implementation of LLMs4Plan is well-detailed, with descriptions of how LLMs are utilized in action selection and candidate set generation.
3. The effectiveness of the proposed framework is empirically validated across ten planning domains, showcasing its practical applicability.

**Weaknesses:**

1. The proposed integration of LLMs into planning frameworks in LLMs4Plan may be complex and difficult to scale.
2. Comparisons with more recent LLM integrated planning baselines is limited.

**Questions:**

1. How does the proposed LLMs4Plan compare with the more recent related work [12] (LLM+P) cited in the paper? And what makes LLMs4Plan a better approach?
2. What potential applications do the authors envision for the LLMs4Plan framework in real-world planning scenarios?

**Limitations:**

yes

---

> ### Author Rebuttal · Authors · 2024-08-07
>
> [Comparison with LLM+P]
> Response #3.1: LLM+P cannot be directly compared because its input and output are different from our LLMs4Plan. The input and ouput of LLMs4Plan are in pddl format, while the input and output of LLM+P are in NLP form. The role of LLMs in LLM+P is more like a kind of semantic understanding and format conversion, i.e., converting NLP problems to pddl problems and use off-the-shelf planners to solve the pddl problems.
>
> We solved the problems from the projects of LLM+P for each of the domains BARMAN, STORAGE, TERMES, and TYREWORLD. The results are as shown below. As we can see, our LLMs4Plan has higher success rate than LLM+P.
>
>  Domain                LLMs4Plan	LLM+P
>
> BARMAN	1.00	1.00
>
> STORAGE	1.00	0.85
>
> TERMES	1.00	0.20
>
> TYREWORLD	1.00	0.90
>
> [Applications to other real-world planning scenarios]
> Response #3.2: For all planning scenarios that graph-based planning can be applied, our LLMs4Plan can also be applied. In addition, with LLMs integrated, our LLMs4Plan is promising on more planning scenarios, e.g., scenarios where domain models are corrupted.

---

### Official Review · Reviewer_Hoc3 · 2024-07-13

**Soundness:** 3
**Presentation:** 3
**Contribution:** 3
**Rating:** 5
**Confidence:** 4

**Summary:**

There have been debates about the fundamental planning abilities of LLMs in planning tasks. To achieve more reliable performance, several recent works have embedded an LLM into a search framework (e.g., MCTS, BFS) and viewed LLMs as heuristics. Along this line, this work take a closer look at the roles LLMs can play in Planning Graph. It considers two tasks for LLMs: pruning actions and sorting actions (as heuristics).

**Strengths:**

- The paper is well-written, with precise language and formalism.
- The experiment is conducted on over 10 domains, making it quite comprehensive.

**Weaknesses:**

1. My biggest concern with this work is that it restricts the use of LLMs to specific roles within a classical planning algorithm. There are many other roles LLMs can play in planning. For instance, see the recent LLM-modulo framework below. Instead of just filtering and ranking actions, LLMs have also been used to evaluate state values or rank plans (i.e., action sequences rather than individual actions).

    - Kambhampati, Subbarao, et al. "Position: LLMs Can't Plan, But Can Help Planning in LLM-Modulo Frameworks." ICML 2024

2. The evaluation based on the number of nodes explored is partial. We should not ignore the time cost (e.g., latency of calling LLMs) + financial cost of using commercial LLMs. It could be very likely that, although LLM+Graph Planning expands fewer nodes, it may take a longer wall-clock time to give the final outputs. I understand that the evaluation could be tricky and it remains an open question for a while. However, the authors should at least make an attempt to address this.

3. In the abstract, this statement is inaccurate: “works have been proposed to investigate the planning effectiveness of LLMs, without considering any utilization of off-the-shelf planning techniques in LLMs.” There have been quite some paper embedding LLMs in off-the-shelf planning algos

    - Zhao, Zirui, Wee Sun Lee, and David Hsu. "Large language models as commonsense knowledge for large-scale task planning." NeurIPS 2023.
    - Yao, Shunyu, et al. "Tree of thoughts: Deliberate problem solving with large language models." NeurIPS 2023.


4. While the corrupted domain model experiment looks interesting, it is unclear what messages it tries to convey. Specifically, why would one run the algo on top of a corrupted domain model when there exists approaches that can leverage LLMs to help complete the domain model before starting the search?

    - Guan, Lin, et al. "Leveraging pre-trained large language models to construct and utilize world models for model-based task planning." NeurIPS 2023
    - Wong, Lionel, et al. "Learning adaptive planning representations with natural language guidance." ICLR 2024.


5. The step of LLM-based action pruning can make the search incomplete, since an LLM may keep ignoring the required action(s) -- in other word, there is no guarantee that the LLM can produce a goal-reaching plan. I notice the authors mention this at a later section (which should be moved to earlier part) that including pruning probabilities could address the problem. I don’t fully agree with this. Can the authors give more detail on how pruning probabilities could guarantee completeness?

6. In the prompt (fig. 3), only the proposition set at the current state is provided. Did the authors consider including the running history of actions (i.e., the partial plan)? Would this affect the overall performance?

7. Line 109: typo in “Algorithm ??”

8. Several works (mentioned earlier) already show that LLMs can be useful heuristics. Can the authors elaborate on the new insights this work provides?

-----
Overall, this study provides a thorough evaluation of LLMs within the Planning Graph algorithm. I appreciate the comprehensiveness of the experiments. However, I also have concerns over the scope of this study (i.e., restricting itself to a limited set of roles). I need to discuss with other reviewers and the authors before finalizing my recommendation.

**Questions:**

See the Weakness section.

**Limitations:**

See the Weakness section.

---

> ### Author Rebuttal · Authors · 2024-08-07
>
> [Restriction in the use of classical planning]
> Response #2.1: Thanks. We think extending our approach to other domains of planning is not an issue that we need to worry about, as any planning domain that can be expressed in natural language form or can be expressed in natural language form through certain transformations is amenable to incorporating LLMs into established planning frameworks. From the perspective of our work, pruning and sorting are very suitable ways of integrating LLMs into GP, a planning framework, at least from the experimental results. In addition, improving classical planning frameworks with LLMs would broaden the applications of classical planning in more real-world scenarios in the planning community.
>
> [Time cost]
> Response #2.2: We are indeed aware of the time cost issue. As we mentioned in the conclusion section of the paper, "the runtime of LLMs4Plan is currently hindered by multiple LLMs calls. While our method requires multiple LLMs calls, it provides substantially improved results. There are also various ways to enhance runtime performance ike using smaller LLMs like Llama [14] or distilling LLMs’ knowledge into a smaller model [13, 7, 11]. Those are interesting avenues for future research."
>
> [Inaccurate statement in the abstract]
> Response #2.3: Thanks for the reminder. We will make the statement more specific, e.g., with respect to deterministic classical planning framework.
>
> [Experiments on corrupted domain models]
> Response #2.4: We agree that there have been approaches aiming at learning domain models automatically, as NeurIPS 2023 and ICRL 2024 papers you mentioned. It is indeed an open challenging problem to investigate effective learning algorithms. There is no doubt that there is still no learning approach that guarantees its learnt domain model is perfect, even though with LLMs. It is still necessitated to explore novel approaches to solve planning problems with corrupted domain models.
>
> [Completeness with respect to the pruning possibility]
> Response #2.5: As we can see from Algorithm 1, when the iteration i (<N) becomes large enough, the pruning possibility \kapa_i will be very small, making Algorithm 1 close to the classic GP algorithm, i.e., no actions are removed with LLMs. In our experiment, we empirically set N to be 7, which can be very large. However, we don't need to make it larger since N=7 is sufficient for our approach to solve all planning problems successfully.

---

> > ### Author Response · Authors · 2024-08-07
> > **Including running history of actions in prompt**
> >
> > [Including running history of actions in prompt]
> > Response #2.6: The history of actions makes the prompt very long, making LLMs suffered from outputing the result. We thus did not consider including history of actions, even though it is possible for improving the overall performance.

---

> > ### Comment · Reviewer_Hoc3 · 2024-08-10
> >
> > I thank the authors for their detailed response. However, I don’t find it convincing enough. Here are some additional notes:
> >
> > - `we mentioned in the conclusion section of the paper, "the runtime of LLMs4Plan is currently hindered by multiple LLMs calls. While our method requires multiple LLMs calls` I don’t think this statement in the conclusion can address my concern over the eval metric. I am aware & most people in relevant communities are aware of the cost of using LLMs. My key point is, a paper like this should not solely use the # of node explored, which is a “partial” metric, to claim advantage.
> > - `LLMs would broaden the applications of classical planning` The issue here is actually twofold. For one, as I just mentioned, without a fair metric, it would be hard to claim whether LLMs broaden the application of classical planning. Secondly, restricting LLMs to certain roles within a classical planning framework will not expand the scope of the problems that the original planning framework can solve (think about limitations like the expressiveness of symbolic domain representation)
> >
> > Overall, I find my original evaluation appropriate for the current manuscript and will therefore maintain the current score.

---

> > > ### Author Response · Authors · 2024-08-11
> > >
> > > Thanks. We would like to clarify that, when we mentioned "broadening the applications of classical planning", we meant application problems with corrupted models that can't be solved by classical planning, may be able to be solved by the integration of LLMs and GP (i.e., our proposed LLMs4Plan), instead of ``broadening'' the expressiveness of symbolic domain representation. --- We are sorry for the confusion. We will make clear of this in the paper.

---

### Official Review · Reviewer_HR5T · 2024-07-29

**Soundness:** 3
**Presentation:** 3
**Contribution:** 3
**Rating:** 6
**Confidence:** 3

**Summary:**

The paper aims to investigate integrating large language models (LLMs) into classical planning frameworks to enhance the planning effectiveness. The authors proposed a novel method named LLMs4Plan which integrates LLMs into action selection and mutual constraints solving within the graph-based planning framework. Evaluated across ten classic planning problems, this approach demonstrates improved success rates and reduced computational complexity compared to traditional methods. The study concludes that while LLMs alone are insufficient for planning, their integration into classical frameworks significantly boosts performance,.

**Strengths:**

1. This paper investigates an intriguing topic: the performance of LLMs in classical planning problems. While the impressive performance of LLMs in natural language processing and coding tasks is well-investigated, their efficacy in planning tasks remains largely unexplored. Understanding whether LLMs can replace classical planning algorithms is a significant and meaningful research question.
2. The paper conducts extensive experiments on ten classical planning problems, which enhances the credibility of its findings and conclusions. This comprehensive evaluation demonstrates the robustness of the proposed approach.
3. The paper reveals that LLMs still cannot surpass classical planning algorithms, thereby highlighting a valuable direction for future research. This insight encourages further investigation into how LLMs can be effectively integrated with traditional planning methods.

**Weaknesses:**

1. Although the authors point out that LLMs cannot outperform classical planning algorithms on their own and need to be integrated with classical methods to perform well, the paper lacks detailed insights on this integration. For example, specific strategies for integrating LLMs with the classic planning algorithms and the roles where LLMs excel within planning problems are not thoroughly discussed. The designed "expandGraph" and "sortActions" may not be the best practice manner. Future research directions to enhance the planning capabilities of LLMs should be more explicitly outlined.
2. The experiments are conducted in simulated planning domains, and the paper does not provide real-world applications or case studies to validate the practical utility of the approach. Including experimental results from more realistic scenarios would strengthen the paper.
3. While the method is effective for graph-based planning, its applicability to other planning frameworks or domains is not thoroughly investigated. A broader analysis could reveal the versatility of the proposed approach.
4. Typos: Algorithm ?? in Line 109.

**Questions:**

What do you foresee as the future of planning algorithm development? Will it be a hybrid approach combining LLMs with classical planning methods, or an end-to-end solution relying solely on LLMs?

**Limitations:**

See the Weaknesses part

---

> ### Author Rebuttal · Authors · 2024-08-07
>
> [Insight of integrating LLMs into graph planning]
> Response #1.1: Thanks. The insight of using LLMs in graph planning is analogous to one of the general ways humans figure out solutions to planning problems, i.e., first looking ahead and then searching back. Given planning problems, human usually conducts two phases to find solutions, i.e., (a) forward building a rough solution scheme starting from initial state s_0 to goal g according to “looking ahead’’ strategies to narrow down the scheme, and (b) “searching back” the scheme according to ``preference’’ strategies of actions. The insights of the “looking ahead” and “preference’’ strategies are implicitly implemented by LLMs corresponding to the two main steps “expandGraphLLMs” and “sortActionsLLMs” in our LLMs4Plan approach. Those two steps correspond to two critical steps that determine the efficacy of graph planning.  It indeed may not be the best practice manner for other planning frameworks, e.g., plan-space based planning, satisfaction-based planning, etc. As mentioned in the last paragraph of the introduction section, in this work we provide new clues for how to deeply embed LLMs into off-the-shelf planners to enhance planning capabilities, i.e., first identifying critical steps in off-the-shelf planners, and then design proper prompt generation to be embedded in to the planners. While it is highly possible that different planning frameworks have their own critical steps, the idea of our LLMs4Plan approach on how to embed LLMs in graph planning can be shared in different planning frameworks. We will add more discussions on future research directions in the conclusion section if the paper is accepted.
>
> [Experiments in real-world applications]
> Response #1.2: Thanks. We agree that conducting experiments in real-world applications is more practical. It is, however, non-trivial to design planning domain models from real-world applications for off-the-shelf planners. On the other hand, the planning domains can indeed be viewed as ones abstracted from real-world domains and used for evaluations of planners in the planning community [12,15].
>
> [Applicability to other planning frameworks]
> Response #1.3: The idea of integrating LLMs into graph planning can be applied to other state and action space search-based planning frameworks.
>
> [Foresee the future of planning algorithm development]
> Response #1.4: We believe this is an open question. Through our related studies, we believe embedding state-of-the-art LLMs into planning frameworks is more promising, compared to solely end-to-end LLMs.
>
>
> [Typos]
> Response #1.5: Thanks. We will revise the typos correspondingly.

---

### Decision · Program_Chairs · 2024-09-25

**Decision:**

Reject

**Comment:**

I decided to write a review. Then, I provide my meta-review

## Review


The paper presents a modification of the known classical-planning algorithm Graphplan that uses LLMs. Graphplan works by creating interleaving layers of actions and facts they can add. LLMs are used to filter those actions beyond the minimal filter of the facts of the preconditions being present in the previous layer. Experiments were performed in 9 known PDDL benchmarks. The experiments compare variations of the algorithm with baselines consisting of GPT-3.5 and GPT-4.

Strengths:
- Proposes a possible integration of LLMs within classical planners.
	- The notion of filtering actions that could lead to a plan might be a suitable way to be incorporated within classical planners.
	- Sorting actions might also be another way, and less risky than filtering.
- Filtering with LLM might enable planning even when actions miss some preconditions or actions.
- Ablation study compare the importance of different components.

Weakness:
- Graphplan is far from being a state-of-the-art classical planner. Comparison with a recent baseline is necessary for understanding how hard are the instances.
	- For instance, Valmeekam et al (NeurIPS 2023), refers to Fast-Downward.
- Only ten instances with such diversity in number of objects might not be informative enough due to high biases. A precise list of instanceswould have helped to clarify this.
- GPT-4 and GPT-3.5 are reported to be random problems between 20 and 40 blocks generated using a known library for generation. However, previous work has reported that as difficult. Here are some previous experiments on the blocks domain.
	- Valmeekam et al. (NeurIPS 2023) do not offer comparable results. However, they report 6.8% in plan generation using Instruct-GPT3 over 600 instances. No plan was found for problems with five blocks.
	- Valmeekam et al, “PlanBench: An Extensible Benchmark for Evaluating Large Language Models on Planning and Reasoning about Change”, (NeurIPS 2023 Track Datasets and Benchmarks homepage, https://openreview.net/forum?id=YXogl4uQUO) reports GPT-4 plan generation success of 34.3% out of 600 problems. The problems feature up to five blocks.
- No specific details are given on using GPT-4 and GPT -3.5 that might explain their surprising performance.
- Number of LLM calls is not reported.

I’ve been following the line of work by Valmeekam et al (NeurIPS 2023), and I have tried myself some simple cases. I’ve used PDDL generators myself.

If we assume that GPT-4 is finding plans for 7 out of 10 blocks world problems with 20 to 40 blocks, some possible explanations are:
- the PDDL generator was used with parameters that make the problems easy.
	- In this case, the comparison between algorithms does not correspond to what’s expected from the domains.
- There is some error when verifying whether the plans are correct.

Given this concern, I’m afraid the paper must be rejected. Moreover, I wonder if NeurIPS is the proper venue for this paper. AAAI, IJCAI, or ICAPS might be better options. Actually, the paper’s scholarship doesn’t refer to the planning literature except for Graphplan and LPG, a planner mentioned by Valmeekam et al. (NeurIPS 2023).

Soundness: 3: good
Presentation: 3: good
Contribution: 3: good
Rating: 3: reject
Confidence: 5: You are absolutely certain about your assessment. You are very familiar with the related work and checked the math/other details carefully.

## Meta review

The reviewers were, in general, positive about the idea of using LLM for improving a classical planning algorithm. There were some concerns about hyper-parameters and comparison with LLM+P, cited in the submission. An important one was completeness, as the LLM filtering might make the problem unsolvable. The rebuttals answer questions about the running time, the number of expanded nodes, comparison with LLM+P, and other uses of LLMs in planners. The reviewers didn’t question the significance of the Graphplan algorithm, or the lack of details on how GPT-4 return correct plans for the planning domains.